# An E2-ubiquitin thioester-driven approach to identify substrates modified with ubiquitin and ubiquitin-like molecules

Gábor Bakos[1], Lu Yu[2], Igor A. Gak[1], Theodoros I. Roumeliotis[2], Dimitris Liakopoulos[3], Jyoti S. Choudhary[2] & Jörg Mansfeld [1]

Covalent modifications of proteins with ubiquitin and ubiquitin-like molecules are instrumental to many biological processes. However, identifying the E3 ligase responsible for these modifications remains a major bottleneck in ubiquitin research. Here, we present an E2-thioester-driven identification (E2~dID) method for the targeted identification of substrates of specific E2 and E3 enzyme pairs. E2~dID exploits the central position of E2-conjugating enzymes in the ubiquitination cascade and provides in vitro generated biotinylated E2~ubiquitin thioester conjugates as the sole source for ubiquitination in extracts. This enables purification and mass spectrometry-based identification of modified proteins under stringent conditions independently of the biological source of the extract. We demonstrate the sensitivity and specificity of E2-dID by identifying and validating substrates of APC/C in human cells. Finally, we perform E2~dID with SUMO in *S. cerevisiae*, showing that this approach can be easily adapted to other ubiquitin-like modifiers and experimental models.

[1] Cell Cycle, Biotechnology Center, Technische Universität Dresden, 01307 Dresden, Germany. [2] Functional Proteomics Group, The Institute of Cancer Research, London SW3 6JB, UK. [3] Centre de Recherche en Biologie cellulaire de Montpellier (CRBM), CNRS UMR 5237, 34293 Montpellier Cedex 05, France. Correspondence and requests for materials should be addressed to J.M.  (email: joerg.mansfeld@tu-dresden.de)

Biological repositories such as BioGRID[1] currently list more than 13,000 posttranslational modifications with ubiquitin and ubiquitin-like molecules (UBLs) on proteins encoded by almost 10,000 genes, indicating that at least half of the proteins encoded in the human genome are modified. Because ubiquitin and UBLs are involved in most cellular processes, it is not surprising that aberrations in ubiquitin and UBL systems have severe consequences such as neurodegenerative diseases or cancer formation[2,3].

Ubiquitination requires the interplay of three enzymes in a highly ordered fashion. First, a ubiquitin-activating enzyme (E1) catalyzes the formation of a ubiquitin thioester on its active cysteine in an ATP-dependent manner. During the second step, ubiquitin is transferred to a ubiquitin-conjugating enzyme (E2) resulting in an E2~ubiquitin thioester (E2~Ub). The E2~Ub subsequently interacts with a ubiquitin ligase (E3), which provides the scaffold for substrate recognition[4]. The majority of E3 ligases contain a really interesting new gene (RING) domain that catalyzes the transfer of ubiquitin from the E2 directly onto a lysine residue of the substrate[5]. In contrast, E3 ligases belonging to the homologous to E6AP C-terminus (HECT)[6] and RING-between-RING (RBR)[7] families initially form an E3~ubiquitin thioester intermediate on a catalytic cysteine before the substrate is modified. Modifications of substrates with UBLs, such as SUMO, FAT10, ISG15, NEDD8, or UFM1 require similar enzymatic cascades, some of which involve enzymes that can recognize more than one modifier[8].

In recent years, advances in mass spectrometry provided not only comprehensive ubiquitome landscapes[9–12], but also increasingly shed light on proteins modified by SUMO[13,14] and other less understood UBLs[15–19]. However, the dynamic and reversible nature of these modifications, the weak and/or transient interaction between ligase and substrate, the significant degree of redundancy and multiplicity between E1, E2, and E3 enzymes, and the rapid destruction of many ubiquitylated proteins still present significant technical challenges in identifying E3 ligase substrates[20]. Current approaches to define enzyme-substrate relations include yeast two-hybrid[21,22], protein microarrays[23,24], substrate trapping[25–28], biotin-dependent proximity labeling (BioID)[29], and engineered ubiquitin enzyme cascades[30]. Alternatively, the abundance of modified substrates can be increased by overexpressing the E3 ligase of interest[31,32]. Given that substrates are targeted for ubiquitin-mediated proteolysis, methods monitoring protein stability in comparison to known model substrates[33] or upon chemical or genetic interference with E3 ligases have been developed as well[34–36]. While each of these approaches has its own strength and weaknesses, we sought a simple and versatile method that can be applied to different experimental models without the need for extensive and time-consuming genetic or protein engineering, and which is independent of the functional outcome triggered by the modification. Here, we demonstrate the specificity and sensitivity of E2 thioester-driven substrate identification (E2~dID) by identifying and characterizing substrates of the anaphase-promoting complex/cyclosome (APC/C) in human cells and of the SUMO E3 ligases Siz1/Siz2 in *S. cerevisiae*.

## Results

### Rationale of E2~dID. 
We reasoned that an affinity labeled modifier, for example, biotinylated ubiquitin (bioUBB), ligated to substrates exclusively by the E3 of choice would enable the straightforward purification and identification of its specific substrates. Further, if the ligation of bioUBB to substrates occurred in vitro in the context of extracts (in extracto) all experimental models that support extract preparation including

tissue cell culture, primary cells derived from multicellular model organisms, and unicellular models, such as yeast, could be used. The biotin tag allows purification of modified substrates under denaturing conditions for subsequent identification by mass spectrometry. E2 and E3-specificity is achieved by chemical inactivation of endogenous E1 and E2 enzymes within extracts and quantitative comparison of substrates identified in the presence or absence of the E3 ligase of interest (Fig. 1a).

### E2~dID with APC/C. 
As a proof of principle, we set up E2~dID in HeLa cells and aimed to identify established mitotic substrates of APC/C. The APC/C is essential for chromosome segregation in eukaryotic cells by targeting CCNB1 and PTTG1 for proteasomal degradation. During mitosis and G1 phase, the E2 enzymes UBE2C and UBE2S are required for initiating and elongating mainly lysine 11-linked ubiquitin chains on APC/C substrates, respectively[37–39]. First, we generated bioUBB by in vitro biotinylation of ubiquitin on a small N-terminal linked AVI-tag[40] (Supplementary Figure 1a). Alternatively, bioUBB can be obtained from commercial sources. To generate E2~bioUBB thioesters that support APC/C-dependent ubiquitination, we combined recombinant human E1 (UBA1), E2 (UBE2C), bioUBB, and ATP in in vitro charging reactions. While UBE2C~bioUBB thioesters were readily formed, UBE2C became auto-ubiquitinated during charging as indicated by a mobility shift of UBE2C on SDS-PAGE that was not sensitive to reduction with dithiothreitol (DTT) (Supplementary Figure 1b). This might decrease the sensitivity of E2~dID as auto-ubiquitinated UBE2C molecules will be purified and identified by subsequent mass spectrometry analysis along with specific APC/C substrates. To reduce auto-ubiquitination, we mutated the conserved lysine 119 located in close proximity to the active site cysteine to arginine (K119R). Indeed, UBE2C$^{K119R}$ exhibited highly reduced auto-ubiquitination (Supplementary Figure 1c), while almost retaining its full activity when used with APC/C to ubiquitinate an N-terminal fragment of CCNB1 (amino acids 1–89) (Supplementary Fig. 1d).

We initially sought to purify UBE2C$^{K119R}$~bioUBB thioesters from charging reactions. However, its short half-life prevented generating sufficient amounts of pure UBE2C$^{K119R}$~bioUBB. Instead, we stopped charging reactions by iodoacetamide (IAA), which alkylates active site cysteines of E1 and E2 enzymes, but does not interfere with pre-formed E2~bioUBB linkages. Importantly, IAA-treated charging reactions supported APC/C-dependent ubiquitination of PTTG1 in vitro, suggesting that IAA does not affect APC/C activity (Supplementary Figure 1e). We also added IAA to anaphase extracts prepared from HeLa cells to preclude endogenous E1, E2, HECT, and RBR E3 enzymes from utilizing the supplied recombinant ubiquitin. Adding 5–10 mM IAA or 50 mM N-ethylmaleinimid (NEM) to extracts was sufficient to prevent conjugation of ubiquitin to proteins (Supplementary Figure 1f) and therefore, we used 10 mM IAA in all subsequent experiments. We decided against NEM because adding NEM, but not IAA at pH = 7.5 inhibited APC/C in vitro (Supplementary Figures 1e, g). Notably, the addition of IAA also inhibits cysteine-containing de-ubiquitinating enzymes and thereby preserves bioUBB-conjugated substrates. Together, this ensured that the E2~bioUBB conjugates supplied with IAA-treated charging reactions act as the exclusive source for bioUBB that is ligated to the substrate by endogenous RING-type E3 ligases in the extract (Fig. 1b). In addition, we treated extracts with the proteasome inhibitor MG132 to prevent potential substrate degradation.

Because many E2 enzymes including UBE2C have been reported to work together with several E3 ligases[41,42], the

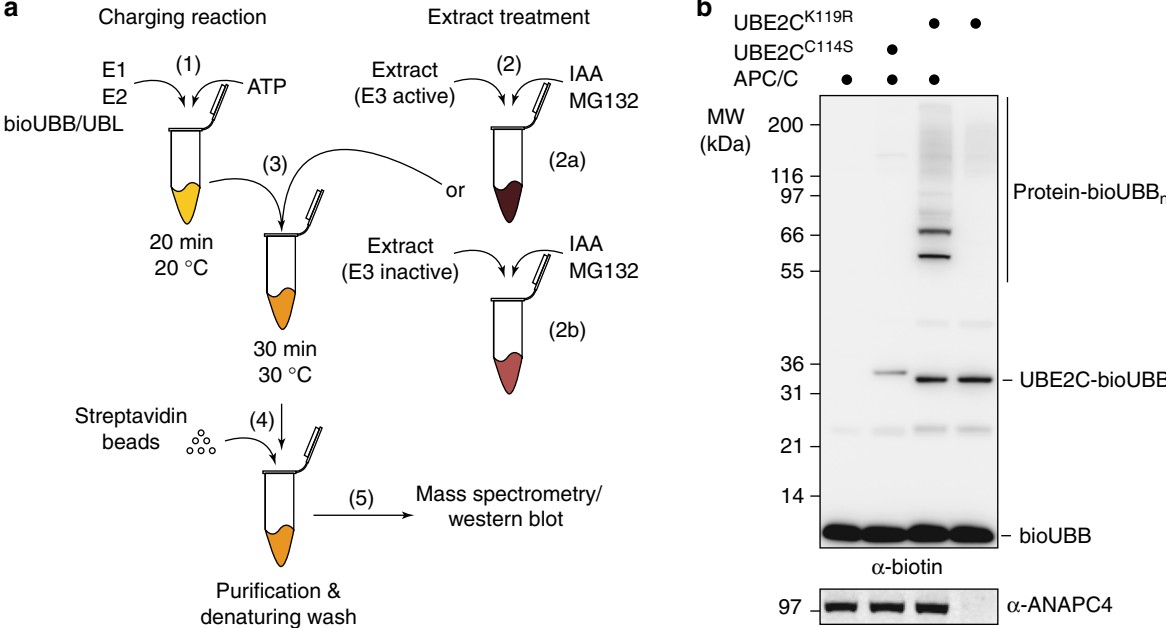

**Fig. 1** Principle of E2~dID. **a** Schematic representation of the E2~dID technique: (1) In vitro generation of E2~modifier conjugates (charging reaction) using biotinylated UBB or UBLs. (2) Cell or tissue extracts are treated with the indicated chemicals to inactivate the endogenous cysteine-dependent ubiquitin and UBL enzymes. To reveal E3-specific substrates extracts with active E3 (2b) or inactive E3 (2b) are prepared. (3) Charging reaction and extracts are combined to initiate modifications in extracto. (4) Modified proteins are purified under denaturing conditions through the biotinylated modifier. (5) Analyses of purified proteins, e.g. by Western blot or mass spectrometry. **b** Representative ($n = 4$) Western blot analysis of E2~dID-dependent labeling of APC/C substrates in extracts with the indicated antibodies (Supplementary Data 7). Note, that ubiquitination of proteins in the extracts depends on the presence of E2~bioUBB conjugates (UBE2C) and ANAPC4 in the extract (compare ± APC/C)

bioUBB-modified proteins might represent substrates from multiple E3 ligases. To reveal to which extent bioUBB modifications depended on APC/C, we performed E2~ID in unperturbed anaphase extracts (E3 active) and anaphase extracts, where APC/C has been removed (E3 inactive) by immuno-depleting its crucial ANAPC4 subunit. ANAPC4 immunoprecipitation also co-depleted catalytically active subunits ANAPC2, ANAPC11, and ANAPC10 (Supplementary Figure 2a) and strongly reduced UBE2C[K119R]~bioUBB-dependent ubiquitination of proteins (Fig. 1b) in agreement with the notion that APC/C is the predominant E3 enzyme that employs UBE2C in mitosis.

Hence, recombinant E2~bioUBB thioesters drive the ubiquitination of proteins in extracts in presence of an alkylating reagent that inactivates the endogenous E1 and E2 ubiquitin enzymes. Therefore, E2~bioUBB thioesters act as the sole source of bioUBB that is conjugated to substrates enabling their enrichment by affinity purification under denaturing conditions.

**E2~dID is E2 and E3-specific.** During E2~dID extracts are supplied with an excess of 24 to 114-fold of recombinant UBE2C[K119R]~bioUBB compared to endogenous UBE2C (Supplementary Figure 2b, c). While this should not affect substrate recognition by E2-compatible E3 ligases, an excess of E2~bioUBB might facilitate atypical E2/E3 pairings resulting in aberrant ubiquitination. To test this scenario, we supplied anaphase extracts with infrared dye-labeled N-terminal CCNB1, either together with UBE2C[K119R]~bioUBB or with Skp1 Cullin Fbox (SCF)-specific UBE2R1~bioUBB. We found that only UBE2C[K119R]~bioUBB supported CCNB1 ubiquitination, but not UBE2R1~bioUBB nor catalytically inactive UBE2C[C114S] (Fig. 2a). Notably, UBE2R1 was active as UBE2R1~bioUBB thioesters were readily formed (Supplementary Figure 2d) and supported the conjugation of bioUBB to proteins in extracts treated with IAA

(Supplementary Figure 2e). We also assayed CCNB1 ubiquitination in the absence of APC/C to exclude that the excess of UBE2C[K119R]~bioUBB drives substrate ubiquitination in an E3-independent manner. This showed that CCNB1 ubiquitination strictly required APC/C (Fig. 2b), suggesting that E3 specificity is retained during E2~dID.

To determine if E2~dID also supports the ubiquitination of endogenous substrates, we performed E2~dID in mock and ANAPC4-depleted anaphase extracts and purified bioUBB-modified substrates using NeutrAvidin beads. Western blot analysis revealed that both, the amount of total ubiquitination and of endogenous CCNB1 required APC/C (Fig. 2c). Thus, E2~dID supports the modification of endogenous substrates in IAA-treated cell extracts in an E2/E3-specific manner.

UBE2C and UBE2R1 are primarily known as APC/C and SCF-associated E2 enzymes, however other E2's are more promiscuous and support ubiquitination by multiple E3 enzymes[41]. Given that E3 specificity is preserved with more promiscuous E2's, this might be of advantage for E2~dID as it circumvents the requirement to know the specific E2-E3 pairing. We tested this idea using UBE2D1, which is known to interact with multiple E3 ligases including APC/C[37,41,43]. Recombinant UBE2D1 readily produced E2~bioUBB thioesters and in contrast to UBE2C did not exhibit extensive auto-ubiquitination (Supplementary Figure 2f). When added to IAA-treated anaphase extracts UBE2D1~bioUBB thioesters efficiently promoted ubiquitination of proteins (Supplementary Figure 2g). As expected UBE2D1~-bioUBB-driven ubiquitination was not as sensitive to APC/C depletion compared to UBE2C[K119R] ~bioUBB thioesters, reflecting its ability to interact with several E3 ligases (compare Supplementary Figure 2g and Fig. 2c). Nevertheless, ubiquitination of CCNB1 required APC/C, indicating that during E2~dID E3 specificity is retained, even with an E2 enzyme supporting multiple E3 ligases (Supplementary Figure 2h).

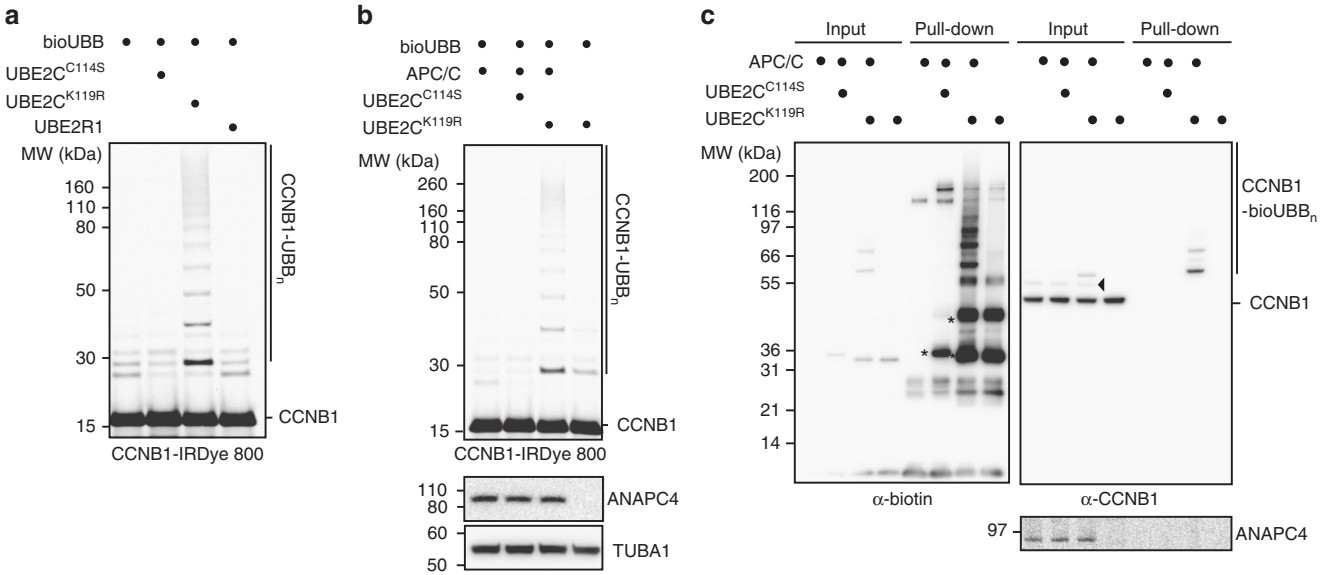

**Fig. 2** E2~dID is E2- and E3-specific. **a** Representative ($n = 4$) SDS-PAGE and fluorescent scanning showing E2~dID in extracts supplied with an IRDye labeled N-terminal fragment of CCNB1. The APC/C-specific E2-conjugating enzyme UBE2C$^{K119R}$, but not inactive UBE2C$^{C114S}$ or the unrelated UBE2R1 supports ubiquitination of the APC/C substrate CCNB1. **b** Representative ($n = 4$) E2~dID analysis as in (**a**) comparing extracts with active E3 and extracts where the APC/C subunit ANAPC4 has been efficiently immuno-depleted as judged by Western blot analysis. **c** Representative ($n = 3$) E2~dID analysis as in (**b**) followed by purification of bioUBB-modified proteins using NeutrAvidin beads and subsequent Western blot analysis. Both total protein ubiquitination (left panel) and specific CCNB1 ubiquitination (right panel) require active UBE2C$^{K119R}$ and APC/C. Note, CCNB1 species modified with endogenous ubiquitin are present in mitotic extracts when APC/C is present (arrowhead). Asterisks indicate auto-ubiquitinated species of UBE2C$^{K119R}$

We conclude that even in the presence of an excess of externally-supplied recombinant E2~bioUBB thioesters E2 and E3 specificity is maintained during E2~dID. Since substrate specificity depends on the E3 ligase, promiscuous E2-conjugating enzymes can be employed for E2~dID as long as they can act together with the E3 of interest.

**Defining the mitotic substrates of APC/C by E2~dID**. Our results confirm that E2~dID enables UBE2C and APC/C-specific ubiquitination of CCNB1 and PTTG1. To assess the performance of E2~dID towards all potential APC/C substrates within extracts, we employed quantitative mass spectrometry based on tandem mass tag labeling (TMT)[44]. Briefly, we performed E2~dID according to Fig. 1a and Fig. 2c, using extracts of cells synchronized in anaphase. We chose mitotic extracts, where APC/C substrates are particularly well-characterized to aid our evaluation of the overall sensitivity and specificity of E2~dID. Notably, we observed a strong Pearson correlation ($R > 0.93$) between individual repeats in all conditions indicative of high reproducibility (Supplementary Fig. 3a). To identify APC/C substrates with high confidence, we accepted only substrates, which exhibited at least a 2-fold ($\log_2 = 1$) enrichment in UBE2C$^{K119R}$~bioUBB reactions compared to control (bioUBB, UBE2C$^{C114S}$) and ANAPC4-depleted (−APC/C) samples (Supplementary Data 1). From the 60 hits that satisfied these criteria 30 have previously been reported, while 26 represent thus far uncharacterized candidates (Fig. 3a and Supplementary Data 2). Except for KATNBL1 and COBLL1 all uncharacterized candidates were identified to be ubiquitinated[1] and 69% contain high-ranking predicted APC/C recognition motifs[45] (Supplementary Data 1). The remaining hits included three APC/C subunits, ANAPC3, ANAPC8, ANAPC13, and the APC/C inhibitor FBXO5. The fact that verified APC/C substrates including PLK1, KIF2C, and ANLN were enriched compared to −APC/C samples, but remained below our threshold (1.75, 1.70, and 1.39-fold enrichment, respectively) suggests that

our dataset contains additional candidates (Supplementary Data 1).

To evaluate the performance of E2~dID in identifying bonafide APC/C substrates, we generated a curated list of 53 well-characterized human and murine mitotic substrates with experimentally-verified APC/C recognition motifs (D box, KEN box, IR/LR tail, ABBA motif)(Supplementary Data 2). Next, we compared E2~dID to three alternative approaches mainly focusing on identifying APC/C substrates during mitosis. First, to co-regulation proteomics, an approach that monitors changes of protein abundance during mitosis by TMT mass spectrometry assuming that candidate substrates share a similar abundance profile with model APC/C substrates[33]. Second, to a mitotic exit ubiquitome based on proteomic snapshots of prometaphase, early and late anaphase cells containing in vivo biotinylated proteins[46]. Third, to antibody-based detection of protein arrays that were incubated with mitotic extracts supplied with an excess of recombinant UBA1, UBE2C, and ubiquitin[23]. Despite suggesting the lowest number of candidates, E2~dID performed best and identified 51% of reference substrates, followed by 34% for the mitotic exit ubiquitome, 32% for co-regulation proteomics, and 21% for protein arrays, respectively (Supplementary Fig. 3b). Notably, with the exception of PLK1, BUB1, and 6 late mitotic exit substrates, which might not yet be targeted by APC/C in the anaphase extracts we used, E2~dID captured the entire set of substrates suggested by the three alternative approaches together (Supplementary Data 2).

Together, this validates E2~dID as a powerful method to identify substrates of E3 ligases based on extracts that can be easily prepared. E2~dID reveals more than half of a reference list containing well-characterized mitotic APC/C substrates and thereby compares well to existing alternative approaches.

**E2~dID candidates are ubiquitinated in vivo**. During E2~dID ubiquitination of proteins occurs in extracts. To assess if substrates suggested by E2~dID are also ubiquitinated by APC/C in

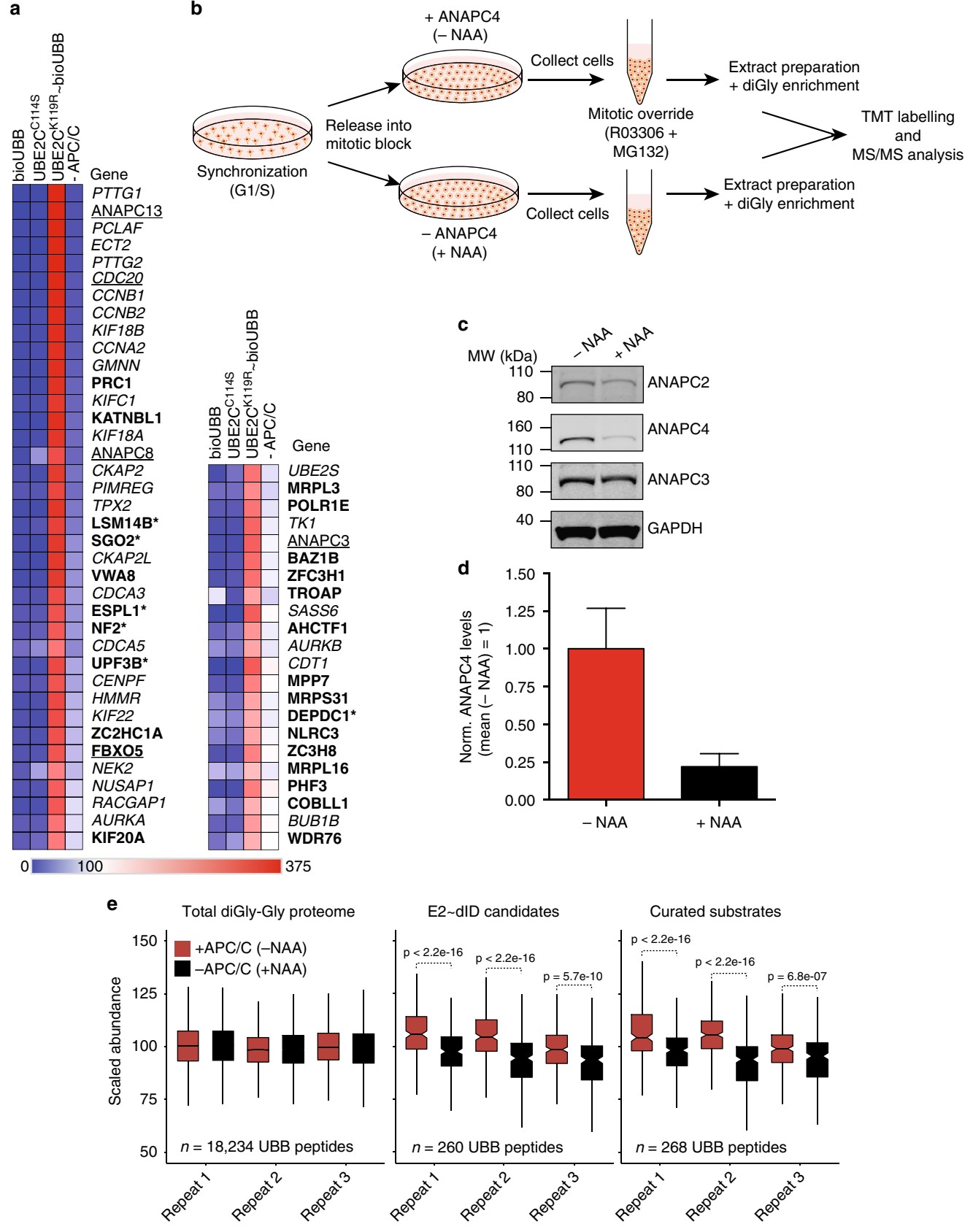

**Fig. 3** Performance of E2~dID with ubiquitin and APC/C. **a** Heat map showing scaled TMT abundances of two independent E2~dID experiments with bioUBB and APC/C performed as shown in Fig. 2c. Only substrates that displayed an at least 2-fold enrichment (E2~bioUBB/− APC/C > 2) are indicated by gene names and are ordered from top to bottom with decreasing enrichment. Italics, known APC/C substrates; underlined, APC/C subunits and regulatory interactors; bold, candidate substrates revealed by E2~dID; asterisks mark candidates selected for further validation. **b** Schematic representation of cell synchronization, APC/C inactivation and sample preparation for diGly MS/MS. (**c**) Representative Western blot ($n = 3$) of cell extracts prepared from − NAA treated (+ ANAPC4) and + NAA treated (− ANAPC4) cells. **d** Quantification of the data shown in (**c**). Bars indicate the mean ± s.e.m. of three independent experiments. **e** Box plot analysis of three independent diGly MS/MS experiments, showing changes in the abundance of ubiquitinated peptides in the presence or absence of APC/C of the total ubiquitinome (left), E2~dID candidates (middle) and curated APC/C substrates (right). Box plots show the median, first and third quartiles and whiskers extend to the smallest or the largest value no more than 1.5-fold of the inter-quartile range. Significance according to the Wilcoxon rank sum test

living cells, we performed quantitative diGly proteomics comparing the mitotic ubiquitinome in the presence and absence of APC/C activity (Fig. 3b). To rapidly inactivate APC/C during mitosis in living cells, we took advantage of auxin-mediated degradation in animal cells[47,48] and the auxin-dependent nanobody mAID-vhhGFP, which targets proteins fused to GFP and GFP-like proteins such as Venus[49]. Adding the synthetic auxin analog 1-naphthaleneacetic acid (NAA) for three hours to prometaphase-arrested cells enabled mAID-vhhGFP to decrease the levels of endogenous 3xFlag-Streptavidin-binding-peptide-Venus-ANAPC4[49] by ~75% (Fig. 3c, d). We have previously shown that this degree of ANAPC4 depletion is sufficient to arrest cells in mitosis and to prevent the destruction of cyclins CCNA2 and CCNB1[49]. Finally, to allow cells proceeding to an anaphase-like state in absence of APC/C activity and protein degradation, we added 10 μM MG132 and 9 μM of the CDK1 inhibitor RO3306 to force mitotic exit (Fig. 3b, see methods).

Quantitative mass spectrometry analyses of + ANAPC4 and − ANAPC4 extracts including diGly-enrichment and TMT-labeling identified more than 18,000 peptides with a diGly signature. 268 peptides were assigned to 38/53 (~ 72%) proteins of curated mitotic APC/C substrates (see above), whereas 260 peptides mapped to 65% (39/60) of candidates suggested by E2~dID (Supplementary Data 3). Notably, the abundance of ubiquitinated peptides of curated substrates and E2~dID candidates were sensitive to ANAPC4 depletion. In contrast, the total mitotic ubiquitinome reflecting substrates of all E3 ligases remained largely unaffected by APC/C inactivation (Fig. 3e).

We conclude that candidate APC/C substrates predicted by E2~dID in vitro are also substrates of APC/C in living cells. Thus, even though E2~dID is performed in vitro its substrate specificity largely recapitulates the situation in vivo.

**Functional analyses of E2~dID candidates in vivo.** E2~dID suggested a high number of candidates that have not been linked to APC/C previously. We selected six substrates covering high (LSM14B and SGO2; 6.61 and 6.11-fold enrichment), medium (ESPL1, NF2, and UPF3B; 5.27, 5.0, and 4.93) and lower (DEPDC1; 2.30) enrichment scores (Supplementary Data 1) for further validation. Whereas ESPL1 is crucial for mitosis and meiosis[50–52], the functions of SGO2 in somatic tissues remain largely unexplored[53]. DEPDC1 regulates mitotic progression[54] and NF2 associates with the mitotic spindle[55]. Together, these candidates appear to fulfill functions in mitosis or meiosis that are typically regulated by APC/C. In contrast, LSM14B and UPF3B have been linked to mRNA turnover, in particular to the CCR4-NOT deadenylase complex[56] and nonsense-mediated mRNA decay[57]—roles the APC/C has not been implicated in thus far.

During mitosis, APC/C ubiquitinates its substrates in an ordered fashion resulting in proteolysis by the 26S proteasome. Thus, we first monitored the stability of selected candidates during mitotic exit by releasing cells from a taxol-induced prometaphase arrest in the presence of the AURKB inhibitor ZM447439 to increase synchronicity within the cell population (see Methods). The levels of SGO2 decreased early during the release and before the anaphase APC/C substrate AURKB[58,59]. In contrast, we did not observe a detectable decrease of LSM14B, UPF3B, and NF2 proteins. Full-length ESPL1 decreased after 30 min resulting in a ~ 170 kDa N-terminal (ESPL1$^{N-term}$) and a ~ 60 kDa C-terminal polypeptide (ESPL1$^{C-term}$), reflecting ESPL1 activation and auto-cleavage once CCNB1 and PTTG1 are degraded. Subsequently, the levels of ESPL1$^{N-term}$ decreased slightly towards the end of the release (105 and 120 min), whereas ESPL1$^{C-term}$ remained largely stable (Fig. 4a). The lack of available antibodies precluded us from assessing the stability of DEPDC1. Nevertheless, during the preparation of this manuscript, DEPDC1 was identified as an APC/C$^{FZR1}$ substrate that is degraded during mitotic exit in a D box-dependent manner[60].

If the decrease of SGO2 and ESPL1 depended on APC/C, then ubiquitination of both candidates should be sensitive to APC/C interference. Therefore, we extracted their ubiquitination state in the presence and absence of APC/C activity from the mitotic diGly ubiquitinome (Fig. 3e). In agreement with APC/C-dependent degradation, the abundances of diGly-linked peptides of SGO2, DEPDC1, and to a lesser extent of ESPL1 were sensitive to ANAPC4 depletion. In contrast, diGly peptides of LSM14B, UPF3B, and NF2 were unaffected (Fig. 4b). To confirm that SGO2 and ESPL1$^{N-term}$ are targeted for destruction by APC/C, we depleted 3xFlag-Streptavidin-binding-peptide-Venus-ANAPC4 with mAID-vhhGFP4 during prometaphase and monitored their stability during a RO3306-induced mitotic override with or without MG132 (Fig. 3b). Depleting ANAPC4 or inhibiting the proteasome stabilized SGO2 similar to CCNB1 or AURKB indicative of APC/C-dependent proteolysis. While ANAPC4 depletion and MG132 treatment delayed ESPL1 auto-cleavage, we did not observe a clear stabilization of ESPL1$^{N-term}$ (Fig. 4c). However, the interpretation of this result is complicated by the observation that ANAPC4 depletion delayed ESPL1 auto-cleavage and thus the generation of ESPL1$^{N-term}$. To confirm SGO2 and ESPL1 as substrates, we assessed their binding to APC/C in interphase (G2 phase) and mitosis (prometaphase). Indeed, SGO2 and ESPL1 predominately co-precipitated with ANAPC3 in mitosis (Fig. 4d). We determined the cell cycle stage of this interaction in more detail by synchronizing HeLa cells into all cell cycle phases (see Methods). Western blotting for cell cycle markers CCNB1 (present in HeLa cells in G2, PM, A, and S), CCNA2 (G2 and S) and CDT1 (PM, A, and G1) confirmed the identity of individual cell cycle stages (Supplementary Fig. 4a). Immunoprecipitating ANAPC3 with a different anti-ANAPC3 antibody indicated that SGO2 and ESPL1 predominately co-precipitated with ANAPC3 from prometaphase to anaphase, thus reflecting the timing of their ubiquitination (Supplementary Fig. 4b).

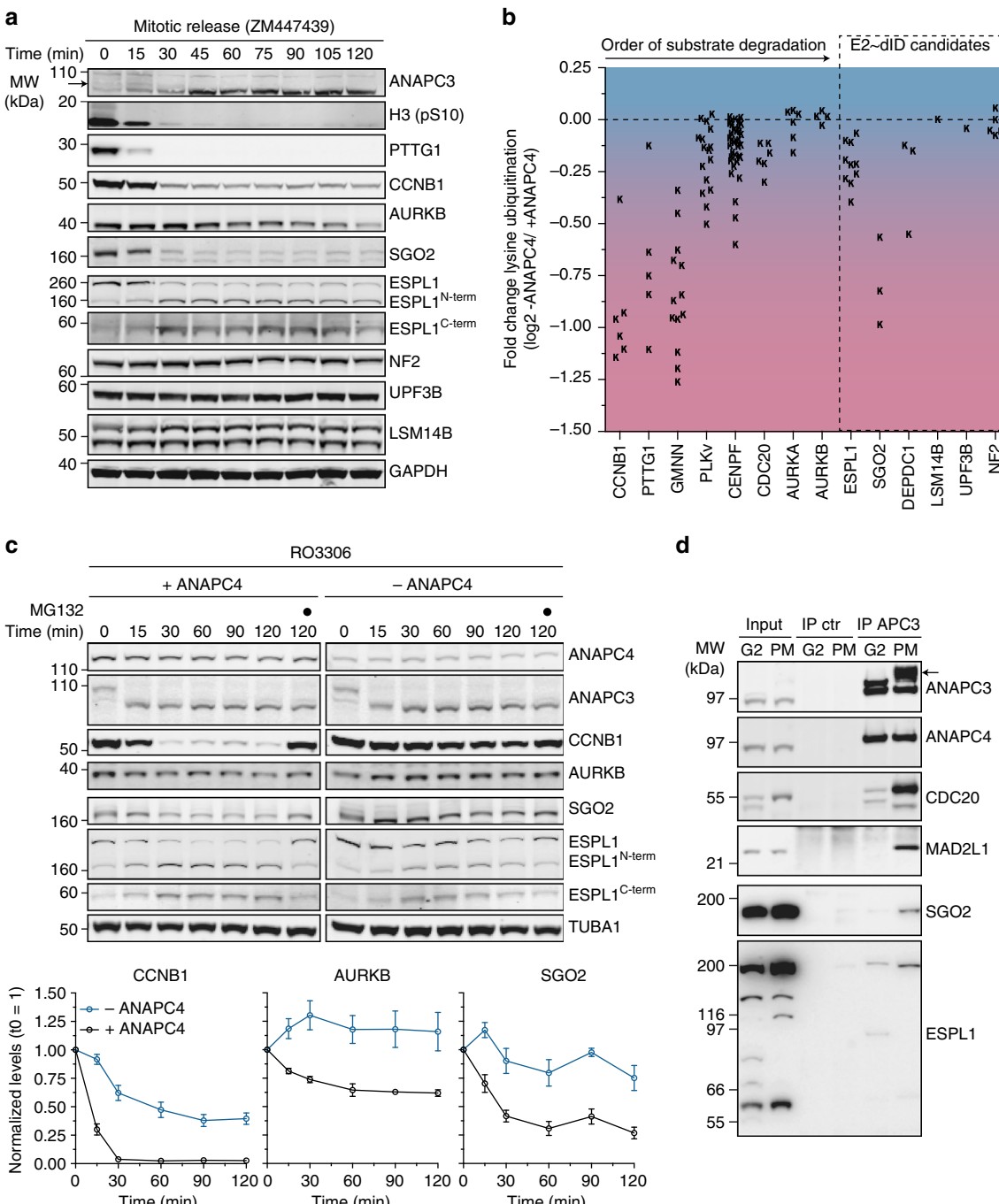

**Fig. 4** Functional validation of APC/C substrates identified by E2~dID. **a** Representative Western blot analysis ($n = 2$) of mitotic release samples using the indicated antibodies. HeLa cells were synchronized in prometaphase with 245 nm taxol (see Methods), collected by mitotic shake-off, washed and released into fresh media containing 10 μm ZM447439. The time-depended decrease in the levels of mitotic markers H3 (pS10), APC/C substrates PTTG1, CCNB1, and AURKB indicate APC/C activation. Arrows indicate phosphorylated ANAPC3. **b** Scatter plot showing the fold-change in lysine ubiquitination in response to ANAPC4 depletion for early APC/C substrates and E2~dID candidates selected for further analyses. Each "K" represents the fold-change (log2) of a ubiquitinated peptide identified from the indicated proteins. The color gradient from blue (no change) to red (negative change) illustrates the fold-change in the abundance of identified ubiquitinated peptides. **c** Representative Western blot analysis ($n = 2$) of a mitotic release for the indicated time-points in the presence ( + ANAPC4) and absence (− ANAPC4) of APC/C activity as described in Fig. 3b, but without the addition of MG132. Note, inhibiting CDK1 by RO3306 is required to allow mitotic exit in absence of CCNB1 ubiquitination and degradation. Quantification of CCNB1, AURKB, and SGO2 levels present the mean ± s.e.m. from two independent experiments with each three technical replicates. **d** Representative Western blot analysis ($n = 3$) of control and ANAPC3 immunoprecipitations from G2-phase (G2) and prometaphase (PM)- synchronized cell extracts showing the interaction of SGO2 and ESPL1 with APC/C. Arrows indicate phosphorylated ANAPC3, while the lower band is derived from prior ANAPC4 detection

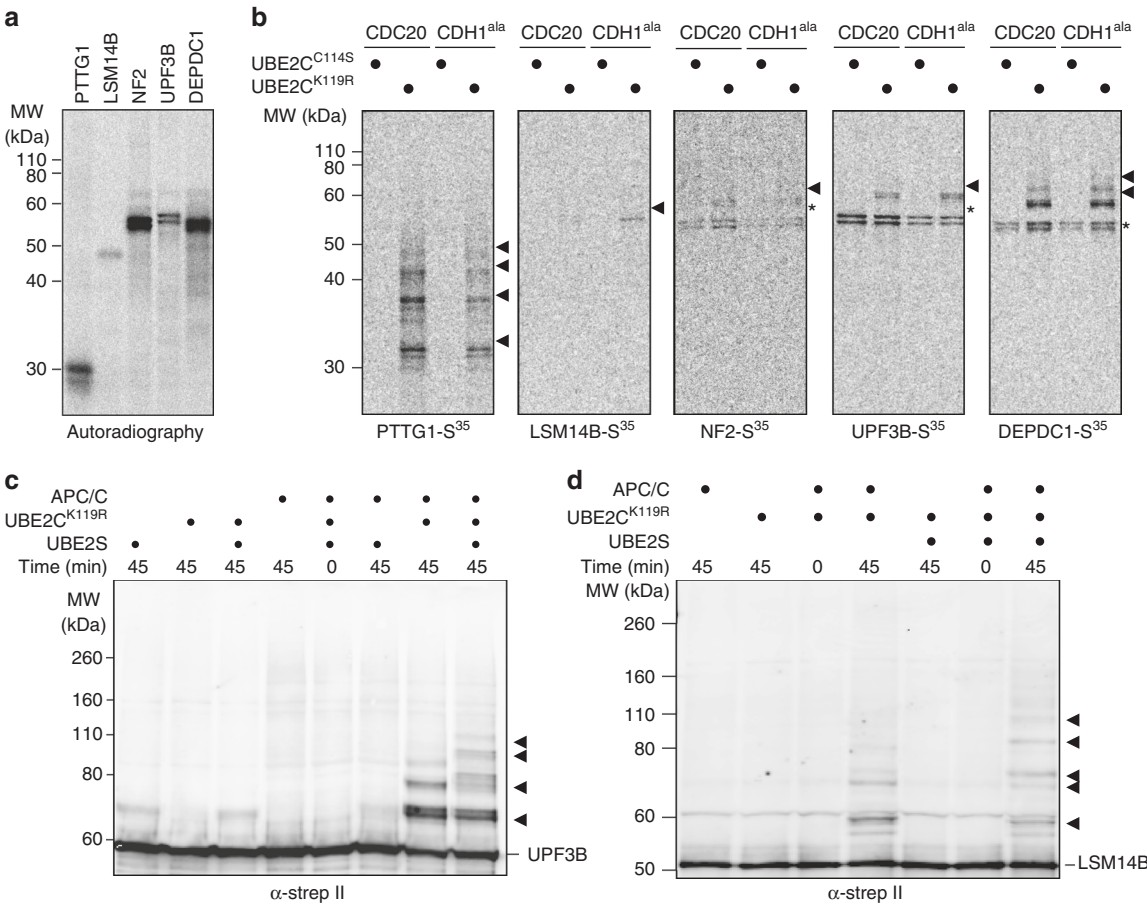

**Fig. 5** UPF3B and LSM14B are substrates of APC/C in vitro. **a** Representative ($n = 7$) autoradiography of methionine S[35]-labelled APC/C candidate substrates used for subsequent E2~dID experiments expressed by coupled in vitro transcription/translation in reticulocytes. **b** Representative ($n = 4$) autoradiography of NeutrAvidin-purified proteins from of E2~dID reactions using CDC20 or FZR1-enriched extracts (see Supplementary Fig. 5a) containing active UBE2C[K119R] or inactive UBE2C[C114S] as well as radiolabeled candidates shown in **a**. Arrowheads indicate bioUBB-modified substrates and asterisks co-purified unmodified substrates. **c** Representative Western blot analysis ($n = 3$) of an in vitro APC/C activity assay with purified components using strep II-tagged UPF3B as a substrate. **d** Representative Western blot analysis ($n = 3$) as in **c** using strep II-tagged LSM14B as a substrate. Note, the addition of ubiquitin chain-elongating UBE2S results in higher molecular weight polyubiquitinated species of UPF3B and LSM14B

Taken together our data suggest that from the selected candidates at least ESPL1, DEPDC1, and SGO2 are ubiquitinated by APC/C in vivo and that ESPL1 and SGO2 interact with APC/C in mitosis. While ubiquitination targets SGO2 and DEPDC1[60] for destruction, its effect on ESPL1[N−term] needs further investigation.

**APC/C ubiquitinates UPF3B, LSM14B, DEPDC1, and NF2 in vitro**. Thus far our experiments in living cells do not provide evidence that LSM14B, UPF3B, and NF2 are bonafide APC/C substrates. Therefore, we attempted to validate UPF3B, LSM14B, and NF2 as APC/C substrates in vitro using a combination of in extracto and APC/C activity assays based on recombinant components. First, we investigated if E2~dID reactions can reveal the covalent attachment of one or more ubiquitin molecules to LSM14B, UPF3B, and NF2. The anaphase extracts used for E2~dID contain APC/C activated by both, CDC20 and FZR1. To uncover a preference for either co-activator we prepared extracts enriched for APC/C[CDC20] and APC/C[FZR1] (Supplementary Figure 5a), added S[35]-labeled in vitro-translated candidates or PTTG1 as a positive control (Fig. 5a), and performed E2~dID. Purification of bioUBB-linked proteins and analyses by autoradiography indicated that PTTG1 and NF2 were preferred

APC/C[CDC20] substrates, whereas LSM14B displayed a preference for APC/C[FZR1]. In contrast, DEPDC1 and UPF3B were ubiquitinated to an equivalent extent by both co-activators (Fig. 5b). To confirm that their modification with bioUBB required APC/C, we repeated E2~dID for each candidate in context of the preferred co-activator in mock or ANAPC4-depleted extracts. Except for NF2, candidate ubiquitination absolutely depended on APC/C (Supplementary Figure 5b). NF2 ubiquitination was also sensitive to APC/C depletion, but was not completely abolished as in case of the other candidates. This residual NF2 ubiquitination may be due to a small (undetectable by Western blot) remainder of APC/C present upon depletion or another E3 ligase working together with UBE2C~bioUBB.

In dividing cells APC/C employs UBE2C as an initiating and UBE2S as a chain-elongating enzyme to assemble lysine 11-linked ubiquitin chains on its substrates[37,38]. However, during E2~dID the IAA-dependent inactivation of all cysteine-containing enzymes prevents chain elongation by endogenous UBE2S. We therefore tested if ubiquitin chains on UPF3B and LSM14B initiated by UBE2C can be elongated by UBE2S in in vitro APC/C activity assays based on purified components. Indeed, the addition of UBE2S triggered poly-ubiquitination of UPF3B (Fig. 5c) and LSM14B (Fig. 5d), as it does for well-characterized APC/C substrates (Supplementary Fig. 3c)[37].

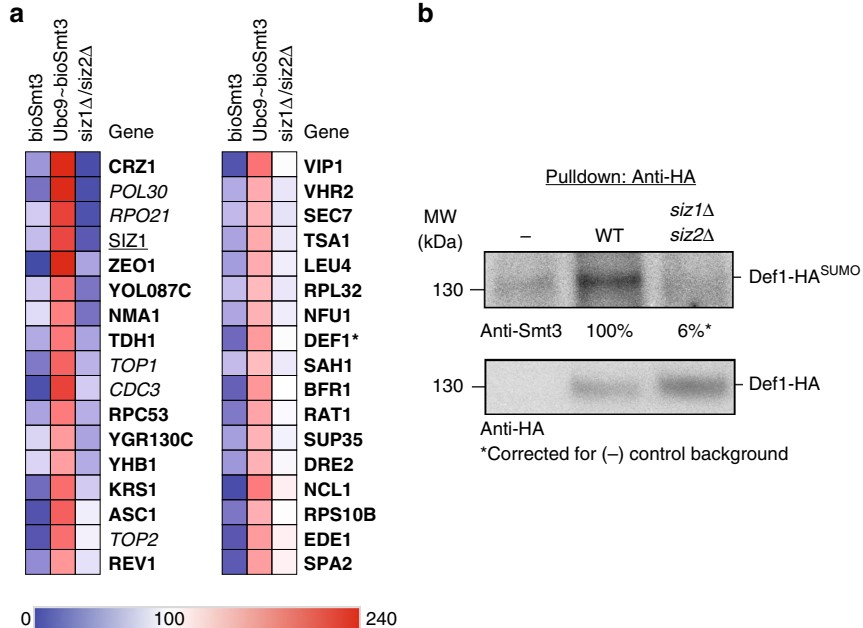

**Fig. 6** E2~dID with SUMO and Siz1/Siz2 in *S. cerevisiae*. **a** Heat map showing scaled TMT abundances of two independent E2~dID experiments with bioSmt3 and Siz1/Siz2 comparing bioSmt3 with Ubc9~bioSmt3 in wild type and *siz1Δsiz2Δ* extracts. Only substrates that displayed an at least 1.41-fold increase (E2~bioSmt3/*siz1Δsiz2Δ* > 1.41) are indicated by gene names and are ordered from top to bottom according to their fold decrease. Italics, known Siz1 and Siz2 substrates; underlined, Siz ligase subunits; bold, candidate substrates revealed by E2~dID; asterisks mark candidates selected for further validation. **b** Representative Western blot (n = 3) analysis of HA immunoprecipitates from WT and *Δsiz1Δsiz2* strains expressing HA-tagged Def1 protein from its endogenous locus

Next, we sought to identify the destruction degrons in UPF3B and LSM14B that are recognized by APC/C. Bioinformatic degron prediction[45] suggested three D boxes and one conserved KEN box as likely candidates for UPF3B (Supplementary Figure 5c, d). We recombinantly expressed variants of UPF3B mutated in each of the three predicted degrons and assessed their abilities to be ubiquitinated by APC/C in vitro. Mutation of the KEN box almost completely abolished UPF3B ubiquitination, whereas mutating either of the D boxes did not have a substantial effect (Supplementary Fig. 5e). LSM14B APC/C-degron prediction identified three conserved motifs within 54 central amino acids of the LSM14B polypeptide: a D box, a classical KEN box and a DEN motif, which acts as a KEN box in *S. cerevisiae* CDC20 and *A. thaliana* PATRONUS during meiosis[61,62] (Supplementary Figure 5f, g). Mutagenesis of the KEN box alone or together with the D box had no effect, whereas combining KEN and DEN mutations prevented LSM14B ubiquitination (Supplementary Figure 5h).

We conclude that in vitro UPF3B, LSM14B, DEPDC1, and NF2 are substrates of APC/C, and like most characterized substrates these can be targeted by APC/C$^{CDC20}$ and/or APC/C$^{FZR1}$. UPF3B and LSM14B ubiquitination is increased by UBE2S and requires KEN motifs as it the case of several APC/C substrates.

**SUMO E2~dID reveals substrates of Siz1/Siz2 in *S. cerevisiae*.** Since other UBLs including SUMO employ similar enzymatic cascades to modify substrates, E2~dID should in principle be applicable to all UBLs that are ligated by E3 enzymes that do not require active site cysteines. Additionally, E2~dID should be independent of the experimental model system as long as cell extracts can be obtained. To test these ideas, we applied E2~dID to wild type and *siz1Δsiz2Δ S. cerevisiae* strains and attempted defining substrates of Siz1 and Siz2 SUMO (Smt3) ligases using

Ubc9 as the E2-conjugating enzyme charged with biotinylated Smt3. During E2~dID with APC/C, we noticed that the two negative controls, bioUBB and UBE2C$^{C114S}$, were highly correlative across experiments (at least $R > 0.93$) (Supplementary Fig. 3c). Therefore, to simplify the workflow of E2~dID, we omitted the catalytically inactive E2 control and processed only bioSmt3, Ubc9~bioSmt3 and *siz1Δsiz2Δ* samples for quantitative mass spectrometry. Similar to E2~dID with ubiquitin and APC/C, we observed a strong correlation (at least $R > 0.91$) between individual repeats and different conditions (Supplementary Fig. 3d). The data were processed as before with the exception that we lowered the threshold for Siz1/Siz2 specificity to 1.41-fold ($\log_2 = 0.5$) (Supplementary Data 4). This accounts for the observation that in yeast Ubc9 can directly modify substrates without the associated E3[63] resulting in partially overlapping substrates.

In total, our study yielded 34 hits that were sensitive to Siz1/Siz2 deletion (Fig. 6a). While more than half (20/34) of identified candidates have been reported to be SUMOylated[1] (Supplementary Data 5) only PCNA (Pol30)[64], septin (Cdc3)[65,66], topoisomerase 1 (Top1)[67], topoisomerase 2 (Top2)[67,68] and RNA polymerase (Rpo21)[69] were previously characterized as Siz1/Siz2-specific substrates (Supplementary Data 5). Thus, investigating the role of SUMOylation of the other candidates might significantly expand our knowledge of Siz1/Siz2 function in yeast.

Taken together, the successful application of E2~dID with ubiquitin and SUMO in human and yeast cells illustrates the sensitivity and versatility of E2~dID and suggests that this approach can be readily extended to other UBLs and experimental models.

**Def1 is SUMOylated in a Siz1/Siz2-specific manner.** To investigate whether E2~dID using Ubc9~bioSmt3 in yeast also suggested Siz1/Siz2 substrates with high confidence we selected the

lower enriched substrate Def1 (1.65-fold enrichment) for further analysis. Def1 acts as a RNA Polymerase II degradation factor during the DNA damage response[70]. We expressed HA-tagged Def1 from its genetic locus in wild type and *siz1Δsiz2Δ* cells, immunoprecipitated the protein from yeast extracts using an antibody against the HA-tag and analyzed its SUMOylation state by anti-Smt3 Western blot. Indeed, precipitates of Def1 were positive for Smt3 in a Siz1/Siz2-dependent manner indicating that it is a Siz1/Siz2 substrate in vivo (Fig. 6b).

Thus, as observed for ubiquitin and APC/C in human cells, E2~dID with Smt3 and Siz1/Siz2 in yeast performs well in identifying E3 substrates.

## Discussion

Identifying substrates of specific E3 ligases remains a major challenge in ubiquitin and UBL biology. Here, we present E2~dID as a versatile and straightforward approach to directly link ubiquitin or UBL-modified substrates to the responsible E2/E3 enzyme pair in a highly sensitive and specific manner. E2~dID relies on cell extracts (Fig. 1) and thus is applicable to any biological source, where sufficient material can be provided. Extracts contain only soluble proteins and may not recapitulate all features required for faithful substrate recognition by E3 ligases, e.g. the contribution of spatial regulation. Nevertheless, key characteristics of the source material such as a particular cell cycle phase, a differentiation stage or tissue-specificity are retained in extracts and will contribute to E3 selectivity and specificity. Indeed, mitotic APC/C substrates suggested by E2~dID in vitro largely overlap with APC/C substrates identified by diGly proteomics in living cells (Fig. 3). Due to the alkylation step E2~dID is only compatible with E3 ligases that do not contain active site cysteines such RING ligases, the by far largest class of E3 enzymes. However, if HECT~UBB, HECT~ISG15, or RBR~UBB conjugates can be produced in vitro, the principle of E2~dID can be readily be extended to these ligase families as well.

Because biotinylated modifiers are provided as in vitro generated E2~modifier conjugates (Fig. 2), there is no need for time-consuming genetic or protein engineering of the source material, which is required for biotinylation approaches in living cells[46,71] or for engineered E1-E2-E3 cascades[30]. Since most E1 and E2 enzymes as well as ubiquitin and UBLs are commercially available or can be easily expressed in *E. coli*, performing E2~dID does not require extensive biochemical expertize. A potential caveat of recombinant E2's, ubiquitin and UBLs is their lack of post-translational modifications that might contribute to function[72–74]. Nevertheless, it is possible that extracts are capable of correctly modifying the supplied recombinant proteins as they are also able to drive ubiquitination using the in the extract contained E3 enzymes.

For increased specificity, E2~dID requires the inactivation of endogenous ubiquitin enzymes, this can be achieved chemically by cysteine alkylation. Here, we have used IAA because in vitro APC/C appears to be sensitive to NEM-treatment (Supplementary Figure 1). IAA can produce lysine adducts that mimic the ubiquitin signature diglycine tag[75]. This is not an issue in the context of TMT labeling experiments because after trypsin digestion the remnant diGly modification contains a primary amine that will react with the TMT label, thus leading to a mass shift that is very different from the mass of lysine adducts. Nevertheless, in other experimental workflows alternative alkylating reagents such as chloroacetamide or bromoacetamide should be considered to avoid misidentification of ubiquitin sites. To increase specificity further, a way to interfere with E3 function is required. Here, we have used antibody-based depletion, auxin-

mediated protein degradation or genetic deletion to interfere with APC/C and Siz1/Siz2, respectively. In addition, the increasing number of small molecule inhibitors targeting ubiquitin and UBL ligases provide attractive alternatives. As substrate specificity depends on E3 ligases, also E2 enzymes that interact with several E3's can be employed for E2~dID, when a quantitative mass spectrometry approach is employed to reveal only substrates of the E3 of interest. We highlight this notion by demonstrating that CCNB1 ubiquitination through UBE2D1~bioUBB thioesters requires APC/C (Supplementary Figure 2), even though UBE2D1 has been suggested to work in addition with at least 15 other E3 ligases[41].

Modification of proteins with ubiquitin or UBLs by endogenous E3 ligases in extracto has previously been applied to substrates immobilized on microarrays[23,24]. However, key differences distinguish these approaches from E2~dID and potentially explain its increased sensitivity in identifying reference APC/C substrates (51% versus 21% or 6%, Supplementary Data 2), and its higher specificity: 45% (27/60) of all candidates suggested by E2~dID contain experimentally validated APC/C recognition degrons[45] compared to only 9% (11/128)[23] or 2% (3/149)[24] of candidates suggested by microarrays. E2~dID takes advantage of endogenous proteins in extracts and is therefore limited primarily by the extent of ubiquitination that can be achieved by the E2~bioUBB conjugate. In contrast, microarrays are limited by the number of immobilized proteins, may sterically occlude the interaction with E3 enzymes, and the recombinant proteins lack native posttranslational modifications that may be required for substrate recognition. Lastly, denaturing washes as employed during E2~dID are required to ensure that only directly modified proteins, but not their modified interactors are identified.

Based on the number of identified reference substrates E2~dID performs also well in predicting bonafide APC/C substrates compared to recent indirect approaches, such as the mitotic exit ubiquitome[46] or co-regulation proteomics[33]. While our study certainly benefited from the ever-increasing sensitivity in mass spectrometry, this is not surprising since both approaches solely depend on the activity profile of APC/C during mitosis to enrich for its substrates. Further, methods that depend on changes in substrate turnover upon interference with E3 ligases of interest[33–36] are only suitable to identify substrates that are targeted for degradation, whereas E2~dID is also compatible with non-degradative modifications.

We provide evidence that six E2~dID-suggested substrates spanning the whole range of our candidate ranking are substrates of APC/C. Ubiquitination of SGO2, DEPDC1, and ESPL1 during mitosis depend on APC/C in vivo (Fig. 4). Indeed, SGO2 and DEPDC1 are destroyed by ubiquitin-mediated proteolysis during mitotic exit, thereby fulfilling key characteristics of typical APC/C substrates (Fig. 4 and[60]). We find that ESPL1 and SGO2 predominately interact with mitotic but not interphase APC/C (Supplementary Figure 4), however unlike for SGO2 we find no evidence that the levels of ESPL1 are sensitive to APC/C inhibition. We note, however, that during G1 phase the levels of full-length ESPL1 and ESPL1^(N−term) are considerably reduced compared to mitosis (Supplementary Figure 4a) suggesting that ESPL1 degradation might occur at a later time-point. APC/C inactivation partially stabilized full-length ESPL1 and delayed the emergence of ESPL1^(N−term) and ESPL1^(C−term) polypeptides (Fig. 4c), presumably because binding of CCNB1 and PTTG1 to ESPL1 hinders its activation even when CDK1 is inhibited. Alternatively, ubiquitination of ESPL1 by APC/C might directly contribute to ESPL1 activation. While we do not find evidence that UPF3B, LSM14B, and NF2 are substrates of APC/C in vivo, at least UPF3B and LSM14B are ubiquitinated by APC/C in vitro, both during in extracto reactions and APC/C activity assays (Fig. 5), and their ubiquitin chains are elongated by UBE2S.

Further, UPF3B and LSM14B ubiquitination requires conserved KEN boxes indicative of bonafide APC/C substrates (Supplementary Figure 5). Hence, why is UPF3B, LSM14B and NF2 ubiquitination not sensitive to ANAPC4 depletion or results in degradation? A likely answer is that the diGly proteomics experiment only provides a snapshot of mitosis during which not all substrates are targeted by APC/C. Indeed, ubiquitination of well-characterized anaphase APC/C substrates AURKA and AURKB was only mildly or not affected by APC/C inactivation, contrary to substrates that are degraded earlier such as CCNB1, PLK1, and CENPF (Fig. 4b). Thus, the mitotic time recapitulated by the diGly ubiquitinome (Supplementary Data 3) presumably reflects early anaphase and therefore cannot reveal APC/C-dependent ubiquitination at later stages of mitosis or during G1 phase. Similarly, the time covered by mitotic release experiments might not be sufficient to reveal changes in the overall protein levels of UPF3B, LSM14B, or NF2. Nevertheless, it is possible that the degree of ANAPC4 depletion is insufficient to stabilize predominately APC/C$^{FZR1}$ substrates that are ubiquitinated towards the end of mitosis or that ubiquitination of UPF3B, LSM14B, and NF2 fulfills non-proteolytic functions.

Compared to ubiquitin E2~dID with APC/C in human cell extracts, SUMO E2~dID with Siz1/Siz2 in *S. cerevisiae* extracts was less sensitive (11%) and specific (18%) in revealing experimentally validated substrates (Supplementary Data 5). We speculate this reflects the ability of Ubc9 to directly SUMOylate substrates without an E3 ligase, albeit with a comparably reduced efficiency[63]. Nevertheless, we show that SUMOylation of Def1 depends on Siz1 and/or Siz2 (Fig. 6), suggesting that the dataset we present here likely contains additional, thus far uncharacterized Siz1/Siz2 substrates.

In summary, E2~dID is a simple, robust and sensitive method for identifying substrates of specific E3 ligases. Based on the conserved enzymology E2~dID is compatible with all UBLs using RING-type E3 ligases and thus should be applicable to a multitude of experimental systems. The functions mediated by ubiquitin and UBLs are essential to any higher organism and aberrations in the interplay between E2/E3 enzymes and their substrates are tightly connected to disease. Hence, E2~dID does not only bear the potential to provide new insights into fundamental cell biological processes, but by virtue of establishing enzyme-substrate relationships may also provide new targets within ubiquitin and UBL systems for therapy.

## Methods

**Molecular cloning.** Constructs for bacterial and mammalian expression were generated by standard molecular biological techniques using the templates, primers and restriction sites indicated in Supplementary Data 6. Inserts of all constructs were verified by sequencing.

**Cell culture and cell synchronization.** Cell lines were cultured according to standard mammalian tissue culture protocol and sterile technique at 37 °C in 5% CO$_2$ and tested in regular intervals for mycoplasma contamination. hTERT RPE-1, HeLa K, and HeLa FRT/TO-3xFlag-FZR1$^{ala}$ cells were a kind gift from Jonathon Pines (ICR, London, UK). HeLa FRT/TO-3xFlag-Venus-SBP-ANAPC4 + TIR1 + mAID-vhhGFP4 cells were described previously[49]. hTERT RPE-1 cells were cultured in DMEM/F12 (Sigma Aldrich) supplemented with 10% (v/v) FBS (Gibco), 1% (v/v) penicillin–streptomycin (Sigma-Aldrich), 1% (v/v) Glutamax (Gibco), 0.5 µg/mL Amphotericin B (Sigma-Aldrich) and sodium bicarbonate (Sigma-Aldrich). HeLa cells were maintained in Advanced DMEM (Gibco) with penicillin–streptomycin (Sigma-Aldrich), Glutamax (Gibco), amphotericin B (Sigma-Aldrich) and supplemented with 2% FBS (Gibco) (HeLa K) or 2% tetracycline free FBS (Gibco) with 200 µg/ml hygromycin for HeLa FRT/TO-3xFlag-FZR1$^{ala}$ cells or 0.5 µg/ml puromycin and 400 µg/ml neomycin for HeLa FRT/TO-3xFlag-Venus-SBP-ANAPC4 + TIR1 + mAID-vhhGFP4 cells.

To obtain mitotic cells extracts HeLa K cells where pre-synchronized at the border of G1 to S-phase by standard single (24 h thymidine) or double thymidine (1$^{st}$ block 19 h, 2nd block 16 h) blocks with 2.5 mM thymidine (Sigma-Aldrich). Then, cells were washed and released into fresh media containing 245 nM taxol (Sigma-Aldrich) or 300 nM nocodazole (Sigma-Aldrich) for 9–15 h until 90–95%

of the cells were arrested in prometaphase due to the action of the spindle assembly checkpoint (SAC). Prometaphase cells were directly harvested by mitotic shake-off. For metaphase (CDC20-enriched cells), prometaphase cells were collected by mitotic shake-off, washed in PBS, re-suspended in media containing 10 µM MG132 (Sigma-Aldrich) and 10 µM reversine (Sigma-Aldrich), and incubated for 60 min on 37 °C to reach metaphase. The addition of reversine inhibits the mitotic checkpoint kinase MPS1 and immediately shuts down the SAC resulting in APC/C$^{CDC20}$ that is free of SAC proteins MAD2L1, BUB1B, and BUB3[76]. For anaphase, prometaphase cells were collected by mitotic shake-off, washed in PBS, re-suspended in media containing 10 µM ZM 447439 (VWR) to inhibit the microtubule tension-sensing arm of the SAC[77], and incubated for 30 min at 37 °C to reach anaphase. To obtain G2- and G1-enriched populations, cells were grown until 50% confluency, pre-synchronized with a double thymidine block (see above), released into fresh media, and collected after 5 h (G2 phase) or 18 h (G1-phase). To obtain extracts enriched for FZR1, HeLa FRT/TO cells were induced to overexpress FZR1$^{ala}$ with 2.5 mM tetracycline (Sigma-Aldrich) for 24 h and harvested by trypsinization. Non-phosphorylatable FZR1-expressing cells arrest in interphase since persistent APC/C activation by FZR1 prevents the accumulation of CCNA2 and CCNB1 and thus prevents mitotic entry.

For diGly proteomics and mitotic releases in the absence of APC/C activity, HeLa FRT/TO-3xFlag-Venus-SBP-ANAPC4 + TIR1 + mAID-vhhGFP4 cells were released from a single thymidine block (see above) into fresh media containing 245 nM taxol for a total of 13 h. After 10 h, half of the dishes were supplied with 500 µM 2-naphthoxyacetic acid (NAA, Sigma-Aldrich) for the last 3 h to induce ANAPC4 degradation. Note, in HeLa FRT/TO cells the residual expression of the mAID-vhhGFP4 in standard FBS is sufficient to induce ANAPC4 degradation. Following NAA treatment both, treated (- ANAPC4) and non-treated ( + ANAPC4) prometaphase cells were collected by mitotic shake-off, washed, re-suspended in fresh media containing 10 µM MG132 and incubated at 37 °C for 30 min to inhibit the proteasome and stabilize ubiquitinated proteins. Subsequently, 9 µM RO-3306 (Sigma-Aldrich) was added for 45 min to induce mitotic exit in the absence APC/C activity and protein degradation. For the mitotic release experiments shown in Fig. 4, no MG132 was added to be able detecting protein degradation and cells were directly lysed in LDS-sample buffer (Thermo-Fisher). In all other cases cells were washed with PBS, pelleted by centrifugation and flash-frozen in liquid nitrogen for later extract preparation.

**Extract preparation.** For extract preparation for E2~dID, cell pellets were re-suspended in 4 ml MEB buffer (30 mM HEPES-NaOH pH = 7.5, 175 mM NaCl, 2.5 mM MgCl$_2$, 1 mM DTT, 10% Glycerol), supplied with 10 µM microcystin (Sigma-Aldrich), 1 mM phenylmethylsulfonyl fluoride (PMSF, Sigma-Aldrich), complete protease inhibitors (Roche) and broken up by nitrogen cavitation[78] in a 4639 Cell Disruption Vessel (Parr Instrument Company). For immunoprecipitation (IP), cell pellets were re-suspended in 1.5 × volume MEB buffer containing 0.25% NP40, supplied with 1 mM PMSF, complete protease inhibitors (Roche), PhosSTOP phosphatase inhibitors (Roche) and incubated on ice for 20 min. Cleared extracts (16,000 × g, 15 min at 4 °C) were stored at −80 °C for up to 2 months. Before in extracto ubiquitination or E2~dID assays, cell extracts were supplied with 10 µM MG132 and 10 mM IAA (Sigma-Aldrich) for 2 h at 4 °C.

**Protein expression and purification.** For cell-free protein expression of PTTG1-StrepII, StrepII-LSM14B, StrepII-UPF3B, StrepII-NF2, and StrepII-DEPDC1, a coupled transcription/translation system (Promega) was used in combination with S$^{35}$-methionine (HARTMANN-Analytic) according to manufacturer's instructions. Following the reaction, each sample was supplied with 10 mM IAA and 10 µM MG132 and incubated for an additional 30 min on ice. Recombinant His- and His-AviTag-UBB, His-UBE2C, His-UBE2R1, His-Ubc9, His-AviTag-*SMT3*, GST-UBA1, N-terminal StrepII-CCNB1 (aa 1–86) and PTTG1-StrepII were expressed in logarithmically growing *E. coli* BL21(DE3) in lysogeny broth (LB) media supplemented with 50 µg/ml kanamycin (Sigma-Aldrich). Expression was induced with 0.5 mM Isopropyl β-D-1-thiogalactopyranoside (IPTG, VWR) at 26 °C for 6 h. Recombinant GST-UBA1, strepII-UPF3B, strepII-LSM14B, and GST-TEV protease were expressed in *E. coli* BL21(DE3) Rosetta 2 in LB media supplemented with 50 µg/ml kanamycin and 35 µg/ml chloramphenicol (Sigma-Aldrich). Expression was induced by 0.3 mM IPTG at 18 °C overnight. Following expression cells were harvested by centrifugation and re-suspended in BEN Buffer (50 mM Tris-HCl pH 8, 250 mM NaCl, 2.5 mM MgCl$_2$, 5 mM β-mercaptoethanol, 10 mM imidazole (Sigma-Aldrich), 1 mM PMSF, 5% glycerol) for Ni-NTA purification, in BEG buffer (50 mM Tris-HCl pH 8, 250 mM NaCl, 2.5 mM MgCl$_2$, 1 mM DTT, 1 mM PMSF, 5% glycerol) for GST purification, in BES buffer (30 mM Tris-HCl pH = 8, 150 mM NaCl, 2.5 mM MgCl$_2$, 1 mM DTT, 0.05% Tween 20, 1 mM PMSF) for StrepII purification, or in BEU buffer (50 mM Tris-HCl pH 8, 150 mM NaCl, 10 mM MgCl$_2$, 0.2 mM DTT, 1 mM PMSF, 5% glycerol) for UBA1 purification. Cells were lysed by sonication and cleared by centrifugation at 50,000 × g on 4 °C for 1 h. His-tagged proteins were immobilized on Ni-NTA (Merck Millipore), washed and eluted in SB buffer (30 mM HEPES-NaOH pH 7.5, 150 mM NaCl, 2.5 mM MgCl$_2$, 1 mM DTT, 10% Glycerol) supplemented with 100 mM, 150 mM, and 250 mM imidazole in a sequential order. Following elution, the proteins were re-buffered into SB buffer without imidazole. GST-tagged proteins, except for GST-UBA1, were bound to Glutathione Sepharose 4B (GE Healthcare)

beads, washed and eluted in 50 mM Tris-HCl pH 8, 100 mM NaCl, 1 mM DTT, 0.5 mM EDTA, 10 mM reduced L-Glutathione, and re-buffered into SB buffer. StrepII-tagged proteins were bound to Strep-Tactin Superflow (IBA Life Sciences), washed, eluted in Buffer E (IBA Life Sciences) and re-buffered into SB buffer. Active GST-UBA1 was purified according to a modified protocol from Hershko and colleagues[79] using bioUBB immobilized on NeutrAvidin beads (Thermo-fisher). Briefly, cleared lysates of cells expressing GST-UBA1 were supplemented with 5 mM ATP and passed on the bioUBB column to covalently link GST-UBA1 to ubiquitin via a thioester bond on its active site. Subsequently, the column was washed sequentially with buffer 1 (50 mM Tris-HCl pH = 8, 10 mM MgCl$_2$, 0.2 mM DTT), buffer 2 (50 mM Tris-HCl pH = 8, 1 M KCl, 0.2 mM DTT) and buffer 3 (50 mM Tris-HCl, pH = 8). Active GST-UBA1 was eluted by 10 mM DTT in EB buffer (50 mM Tris-HCl pH = 8, 10% Glycerol) and re-buffered into SB buffer. Purified N-terminal CCNB1 and PTTG1 were labeled by IRDye 680RD or IRDye 800CW maleimides (Li-Cor) according to the manufacturers' instructions. Recombinant UBE2D1 was a kind gift of Jonathon Pines (ICR London, UK).

**UBB and SMT3 biotinylation.** For in vitro biotinylation purified His-AviTag-UBB and His-AviTag-SMT3$^{KGG}$ were re-buffered into BRB buffer (50 mM bicine-NaOH pH = 8.3, 10 mM Mg(OAc)$_2$). Then 40 µM His-AviTag-UBB or His-AviTag-SMT3 were mixed together with biotin (10 mM), GST-BIRA (4.5 µg enzyme/1 nmol UBB) and ATP (10 mM) in BRB buffer and incubated for 3 h at 30 °C. Following the reaction biotinylated His-AviTag-UBB and His-AviTag-SMT3 were purified from the reaction mix via the His-tag under standard Ni-NTA purification conditions (see Supplementary Figure 1).

**E2 charging reactions.** For E2 charging, recombinant GST-UBA1 (280 nM), UBE2R1 (8.5 µM), UBE2D1 (13.73 µM) or UBE2C$^{K119R}$ or UBE2C$^{C114S}$ (11.5 µM) and bioUBB (8.30 µM) were mixed together in RBU (50 mM HEPES pH = 7.5, 100 mM KCl, 2.5 mM MgCl$_2$) in a 25 µl reaction, pre-incubated for 15 min at room temperature (RT). Charging was initiated by the addition of 2 mM ATP and incubation for 20 min at 20 °C. For Ubc9 charging, 100 nM AOS1-UBA2 (Boston Biochem), Ubc9 WT (12.75 µM) and bioSMT3 (8.2 µM) were mixed together in RBS (20 mM HEPES pH = 7.5, 110 mM KCl, 2.5 mM MgCl$_2$) in a 25 µl reaction, pre-incubated for 15 min at RT. Charging was initiated by the addition of 5 mM ATP and incubation for 35 min at 30 °C. For SDS-PAGE analyses reactions were stopped by LDS-sample buffer without DTT or with 50 mM DTT and the DTT containing samples were boiled at 95 °C for 5 min. For subsequent in extracto or E2~dID assays charging reactions were treated with 10 mM fresh IAA for 30 min on ice before use.

**In vitro ubiquitination assays.** APC/C was purified from extracts using mono-clonal anti-ANAPC4 or anti-HA antibodies (see Supplementary Data 7 for dilu-tions and catalogue numbers of all antibodies used in this study). Briefly 400 µg antibodies were coupled to 1 ml Protein G Dynabeads (Thermo-Fisher) according to the manufacturer's instructions and equilibrated in MEB buffer. For in vitro ubiquitination assays, ANAPC4 antibody-beads were mixed with cell extracts at a 1 µg antibody to 166 µg extract ratio for 2 h on 4 °C on a wheel. Subsequently, beads were washed 3× with MEB buffer and 2× with URB buffer (30 mM HEPES-NaOH pH 7.5, 175 mM NaCl, 6 mM MgCl$_2$, 0.05% Tween-20, 1 mM DTT, 5% Glycerol), re-suspended in URB, and 2–3 µg antibody-beads were combined with enzymes and substrates in 13–15 µl reactions. A reaction contained 46.5–53.76 nM GST-UBA1, 341.9–394.5 nM UBE2C WT or K119R, 115.9–133.8 nM UBE2S, 36.7–42.4 µM His-UBB, 2–2.3 mM ATP and 10.1–11.7 mM Phosphocreatine (PC), 2.3–2.6 µM creatine kinase (CK) and 1 µM bovine serum albumin (BSA). For detection, substrates were either IRDye-labelled (see above) or StrepII-tagged for Western blot analysis. Reactions were incubated for 45 min at 30 °C, stopped by the addition of LDS-sample buffer (Thermo-Fisher) containing 50 mM DTT, followed by separation of proteins on SDS-PAGE and detection of fluorescently-labeled substrates by infrared scanning (Li-Cor) or Western blot using the indicated antibodies (see Supplementary Data 7 for dilutions and catalogue numbers). For in vitro ubiquitination reactions shown in Supplementary Figure 1, UBE2C char-ging reactions (see above) treated with 10 mM IAA were used as a source of UBE2C~ubiquitin conjugates.

**In extracto ubiquitination assays and E2~dID.** To deplete APC/C from extracts ANAPC4 or HA (mock treatment) antibody-beads were mixed with cell extracts (10 µg antibody/ 1000 µg extract) and incubated for 30 min at 4 °C on a wheel. To ensure efficient depletion, the process was repeated 3× more times with fresh antibody-beads. Mock or ANAPC4-depleted extracts were supplemented with fluorescently-labeled N-terminal CCNB1 IRDye-680, charging reactions (see above) in a 1 reaction/60 µg extract ratio, incubated for 30 min at 30 °C, stopped by the addition of LDS-sample buffer containing 50 mM DTT and analyzed by SDS-PAGE and fluorescent infrared scanning (Li-Cor). Ubiquintinated proteins from in extracto assays with $^{35}$S-methionin (1 charging reaction/100 µg extract) labeled PTTG1, LSM14B, NF2, UPF3B, and DEPDC1 were purified from reactions with Dynabeads MyOne Streptavidin T1 beads (Thermofisher), washed 2× with buffer 1 (100 mM Tris-HCl pH = 8, 8 M urea), 1× with buffer 2 (100 mM Tris-HCl pH = 8, 8 M urea) and 2× with RIPA buffer (25 mM Tris-HCl pH = 8, 150 mM NaCl, 0.1%

sodium dodecyl sulfate, 0.5% sodium dodecyl sulfate (SDS), 1% Triton X-100), eluted with LDS-sample buffer containing 50 mM DTT at 96 °C for 10 min and analyzed by SDS-PAGE and autoradiography on a phosphorimager (FUJIFILM, FLA-3000). E2~dID assays processed for analysis by mass spectrometry were performed as described above but scaled up (10 mg extract per condition at 10 mg/ml supplied with 35 charging reactions, which is equivalent to a 24.4-fold excess of recombinant UBE2C$^{K119R/CS}$ over the endogenous UBE2C, (see Supplementary Figure 2b). After incubation at 30 °C for 30 min 100 µl of equilibrated NeutrAvidin beads were added, followed by incubation for 30–45 min at 4 °C on a wheel, 3 × washing with WB1 (100 mM Tris-HCl pH = 8, 8 M Urea, 0.5% SDS), 3 × washing with WB2 (100 mM Tris-HCl pH = 8, 4 M Urea, 0.5% SDS), 3x washing WB3 (D-PBS, 0.5% SDS), 3 × washing with WB4 (100 mM Triethylammonium bicarbonate (TEAB) pH = 8.5, 8 M Urea), 2× washing with WB5 (100 mM TEAB pH = 8.5, 4 M Urea) and 2 × washing with WB6 (100 mM TEAB pH = 8.5). For Western blot analysis, the beads were eluted with LDS-sample buffer containing 50 mM DTT and boiled at 96 °C for 10 min. For mass spectrometry (MS) analysis, the beads were re-suspended in an equal volume of digestion buffer (100 mM TEAB pH = 8.5, 10 mM tris(2-carboxyethyl)phosphine (TCEP), 10 mM IAA), incubated at RT for 60 min, supplied with 2 µg MS grade Trypsin (Thermofisher) and incubated overnight at 37 °C. The digested peptides were filtered in C8 filter-tip (in-house made) to remove possible magnetic beads, dried in a SpeedVac and labelled with TMT10plex as instructed by the manufacturer (Thermo Fisher). Labeled peptides were mixed, dried in a SpeedVac and fractionated on a U3000 HPLC system (Thermo Fisher) using an XBridge BEH C18 column (2.1 mm id × 15 cm, 130 Å, 3.5 µm, Waters) at pH = 10, and a flow rate at 200 µl/min in 30 min linear gradient from 5–35% acetonitrile/NH$_4$OH. The fractions were collected at every 30 sec into a 96-well plate by columns, concatenated by rows to 8 pooled fractions and dried in the SpeedVac.

**Co-immunoprecipitation and western blotting.** APC/C was immunoprecipitated (IP) from extracts using custom mouse monoclonal ANAPC3 (Supplementary Figure 5b) or commercial (Bethyl) rabbit polyclonal anti-ANAPC3 antibodies (Fig. 4d). Control (ctr) immunoprecipitations (IP) were performed with mouse monoclonal anti-FLAG or non-specific rabbit IgG antibodies (see Supplementary Data 7 for dilutions and catalogue numbers). Briefly 400 µg antibodies were cou-pled and crosslinked to 1 ml Protein G Dynabeads (Thermo-Fisher) according to the manufacturer's instructions. For IP, Flag and ANAPC3 antibody-beads were equilibrated in MEB buffer and added to cell extracts at a 16 µg antibody to 1000 µg extract ratio and incubated for 2 h 4 °C on a wheel. Subsequently, beads were washed 6× with MEB buffer and then re-suspended in 1X LDS-sample buffer. For IPs with rabbit polyclonal anti-ANAPC3 antibodies, 1 µg ctr or ANAPC3 anti-bodies were added to 1000 µg extract for 1 h at 4 °C on a wheel. Then 10 µl MEB-equilibrated Protein G Dynabeads were added for 1 h, followed by 6 washes in MEB buffer and resuspension in 1× LDS-sample buffer. Samples were eluted from beads by boiling at 65 °C for 5 min, separated by SDS-PAGE on Bis-Tris 4–12% gradient gels in MES or MOPS buffer. Western blotting was performed in 20% ethanol/MOPS buffer (all Thermo Fisher Scientific) in a Mini Trans-Blot electro-phoretic cell (Bio-Rad), and detected with the indicated antibodies (see Supple-mentary Data 7 for dilutions and catalogue numbers). Uncropped scans of blots presented in the main figures are provided in Supplementary Figure 6 within the Supplementary Information.

**Yeast methods and Def1-HA immunoprecipitation.** Yeast Media and genetic manipulations were done using standard methods. The Def1-HA strain has the S288C background. Anti-HA immunoprecipitations were performed using glass-bead lysates from 100 OD yeast cells grown in YPD, in a buffer containing 50 mM TrisCl pH 7.6. 150 mM NaCl, 1 mM MgCl$_2$, 1 mM EGTA, supplemented with 1 mM PMSF, yeast protease inhibitor cocktail (Sigma P8215)) and 25 mM NEM for 2 h at 4 °C using the mouse anti-HA antibody (Sigma HA-7, H3663) and Pan-mouse Dynabeads (Invitrogen). Beads were washed 4× with the same buffer at RT and once with 105 NH$_4$OH solution pH = 11 for 30 min at 37 °C. HA-Def1 was finally eluted by adding 2× Laemmli buffer and heating for 10 min at 90 °C. Western blots were performed using the same mouse HA-7 antibody or the anti-Smt3 antibody (gift from F. Melchior).

**Ubiquitin remnant peptide enrichment.** Cell pellet was lysed in 5 mM IAA in 5% SDS/100 mM TEAB, and processed by ultrasonic probe and heated at 90 °C for 10 min, then processed again by ultrasonic probe. Lysate was cleared by centrifuge at 16,000 × g for 15 min. Protein concentration was measured by Pierce 660 nm Protein Assay (Thermo), then aliquoted at 5 mg each. After reduced by 10 mM TCEP at 56 °C for 15 min and alkylated by 10 mM IAA at RT for 30 min, proteins were precipitated by chloroform/methanol. 1 ml of 100 mM TEAB was added to the protein pellet and the mixture was left in ultrasonic bath for up to 30 s to disperse the pellet well. For two of three 5 mg-aliquot of each sample, 20 µg of Lys-C (Wako) was added and digested for 2 h at 30 °C. Then to all 5 mg-aliquot samples, 80 µg trypsin (Pierce) was added and incubated at 37 °C for 2 h. Another 80 µg trypsin was added and incubated for further 15 h. The digest was then heated at 70 °C for 10 min then dried in SpeedVac.

For ubiquitin-peptide enrichment, 15 mg of the above digest was taken for each condition ( + APC/C and −APC/C): two of 5 mg-aliquot with Lys-C pre-treated, and one without. (Based on the charge distribution map of the LC-MS/MS analysis, the digestion efficiency was the same with or without Lys-C pre-treatment.) We analyzed three replicates for each condition (6 samples in total). The enrichment was performed in three consecutive IP steps: twice with PTMScan® Ubiquitin branch motif (K-ε-GG) immunoaffinity beads (Catalogue #5562) from Cell Signalling Technology (CST), followed by enrichment with the GX41 antibody (Lucerna Technologies). Briefly, in the 1st IP, 1500 μl IAP buffer (50 mM MOPS (pH = 7.2), 10 mM sodium phosphate and 50 mM NaCl, CST) was added to each sample to dissolve the peptides with the assistance of ultrasonic bath up to 30 s, and then the mixture was centrifuged at $16,000 \times g$ to remove any precipitation. To the cleared peptide digest, 1 vial of PBS pre-washed ($4 \times$) CST antibody-beads were added, and incubated with rotation for 2 h at room temperature (RT). The mixture was centrifuged at $2000 \times g$ for 30 s and the supernatant was collected to perform the 2nd IP by incubation with another vial of the CST antibody-beads overnight at 4 °C, and then at RT for another 2 h, and centrifuged as above. Supernatant was collected again, and 100 μg of GX41 antibody and 150 μl of Protein G-Dynabeads (Thermo) were added and incubated at RT for 2 h before the supernatant was discarded. Each IP's ubiquitin-peptide enriched beads ($2\times$ by CST and $1\times$ GX41) were washed twice with IAP buffer, twice with PBS and once with cold HPLC water, then incubated twice with 55 μl of 0.15% TFA for 10 min to eluted the ubiquitin-peptides. The eluates from the same sample were pooled and desalted in a home-made SDB-XC (Empore™, 3 M) tips to remove antibody in the sample, and then dried in SpeedVac, labeled by TMT10plex, pooled and SpeedVac dried again before HpH fractionation and concatenated to 8 fractions.

**LC-MS/MS analysis**. For peptides from E2~dID experiment, the LC-MS/MS analysis were performed on the Orbitrap Fusion Tribrid mass spectrometer coupled with U3000 RSLCnano UHPLC system. Both instrument and columns used below are from Thermo Fisher. The peptides were first loaded to a PepMap C18 trap (100 μm i.d. × 20 mm, 100 Å, 5 μm) for 10 min at 10 μl/min with 0.1% FA/H₂O, then separated on a PepMap C18 column (75 μm i.d. × 500 mm, 100 Å, 2 μm) at 300 nl/min and a linear gradient of 4–28% ACN/0.1%FA in 180 min/cycle at 210 min for each fraction. The data acquisition used the SPS10-MS3 method with Top Speed at 3 s per cycle time. The full MS scans (m/z 380–1500) were acquired at 120,000 resolution at m/z 200 with a lock mass at 445.12003, and the AGC was set at 4e5 with 50 ms maximum injection time. Then the most abundant multiply-charge ions ($z = 2–6$, above 5000 counts) were subjected to MS/MS fragmentation by CID (35% CE) and detected in ion trap for peptide identification. The isolation window by quadrupole was set m/z 1.0, and AGC at 1e4 with 35 ms maximum injection time. The dynamic exclusion window was set ± 10 ppm with a duration at 40 s, and only single charge status per precursor was fragmented. Following each MS2, the 10-notch MS3 was performed on the top 10 most abundant fragments isolated by Synchronous Precursor Selection (SPS). The precursors were fragmented by HCD at 65% CE then detected in Orbitrap at m/z 100–500 with 50 K resolution to for peptide quantification data. The AGC was set 1e5 with maximum injection time at 105 ms.

Ubiquitin remnant peptide enrichment analyses were performed on a Orbitrap Fusion Lumos, and the analysis method was MS2 based at very similar conditions to the above but with several modifications: The gradient was 90 min/120 min cycle. Peptide ions with charge at 3–6 were selected for HCD fragmentation at 38% collision energy. The isolation width was set 0.7 Da. The fragment ions were detected in Orbitrap with 30 K resolution, and the AGC was set 50,000 with maximum injection time at 100 ms.

**Data analysis**. The LC-MS/MS data were processed in Proteome Discoverer 2.1 (Thermo Fisher Scientific) using the SequestHT search engine to search against the reviewed Uniprot protein database of *Homo sapiens* (20,238 entries, Swiss-prot), or *Saccharomyces cerevisiae* (7904 entries, Swiss-Prot), plus the in-house contaminate database. The precursor mass tolerance was set at 20 ppm and the fragment ion mass tolerance was set at 0.5 Da. Spectra were searched for fully tryptic peptides with maximum 2 miss-cleavages. Carbamidomethyl at Cys was set as static modifications, and the dynamic modifications included N-acetylation (Protein N-terminus), Deamidation (N, Q), Oxidation (M), TMT6plex (Peptide N-terminus, K), and GGTMT6plex ( + 343.2059) (K). Note, we used a SMT3$^{KGG}$ mutant[13] to obtain a shorter SUMO remnant. Peptides were validated by Percolator with $q$-value set at 0.05 for the Decoy database search. The search result was filtered by the Consensus step where the protein FDR was set at 0.01 (strict) and 0.05 (relaxed). The TMT10plex reporter ion quantifier used 20 ppm integration tolerance on the most confident centroid peak at the MS3 level. Both unique and razor peptides were used for quantification. Peptides with average reported S/N > 3 were used for protein quantification. Only master proteins were reported (Supplementary Data 1, UBE2C E2~dID and Supplementary Data 4, Ubc9 E2~dID).

Scaled TMT abundances were used to generate heat maps (Morpheus, Broad Institute, USA) and calculate thresholds as indicated in the main text and the legends of Supplementary Data 1 and 4. Modification with ubiquitin and SUMO according to BioGRID[1] and APC/C degrons prediction according to ProViz[45] accepting only degrons with a disorder score of > 0.5. Reproducibility between data sets was determined according to Pearson correlation calculated by R. Visualization

of data as a Venn diagram was done with InteractiVenn[80]. To generate a unbiased reference list to compare the sensitivity and specificity of E2~dID with alternative approaches, the mitotic and mitotic exit substrates (human and mouse and their isoforms) of APC/C listed at ProViz[45] were extracted and only substrates with degrons verified by mutation or deletions experiments were accepted (Supplementary Data 2, including references). Candidate substrates suggested by co-regulation proteomics[33] were compiled according to the authors thresholding (1st percentile) combining the hits of reference clusters 1–6. Candidate substrates suggested by the mitotic exit proteome[46] included all proteins designated as C- and M-specific. Candidate substrates suggested by protein microarrays[23,24] included all proteins designated by the authors as positive hits. The reference list of experimentally-verified Siz1 and Siz2 substrates was assembled from the literature (Supplementary Data 5, including references). Sensitivity is defined by how many candidates were identified from the reference list (in %). Specificity is defined by how many of all hits proposed by each study are part of the reference list (in %).

**Statistical methods**. Prism 6.0 (Graphpad) and RStudio were used for statistics and to create graphs. All data are representative of at least three independent repeats if not otherwise stated. The notation $n$ refers to the number independently performed experiments representative of the data shown in the figures. Significance of the data shown in Fig. 3e was determined by a Wilcoxon rank sum test. No randomization or blinding was used in this study.

## Data availability

Generated plasmids and cell lines are available from the corresponding author upon request. The mass spectrometry proteomics data have been deposited to the ProteomeXchange Consortium via the PRIDE partner repository with the dataset identifier PXD008624. All other data supporting the findings of this study are available from the corresponding author on reasonable request.

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

## Acknowledgements

J.M. is supported by the German Research Foundation (DFG) (Emmy Noether; MA 5831/1–1) and receives funding from the European Research Council (ERC) under the European Union's Horizon 2020 research and innovation program (grant agreement no. 680042). G.B. and I.A.G were members of the Dresden International Graduate School for Biomedicine and Bioengineering (DIGS-BB) PhD program, G.B. received a wrap up DIGS-BB Postdoc fellowship and I.A.G. was funded by a DIGS-BB PhD fellowship. J.S.C, L.Y., and T.I.R. were funded by grant from the Wellcome Trust (WT098051). We are grateful to Magdalena Gonciarz and Doris Müller for technical support, Frauke Melchior for anti-SMT3 antibodies, Nils Kröger for excellent technical input, Sonja Lorenz for critically reading the manuscript and Olaf Stemmann for SGO2 and ESPL1 antibodies and the sharing of unpublished results. We acknowledge support by the Open Access Publication Funds of the SLUB/TU Dresden.

## Author contributions

Conceptualization, J.M.; Methodology, G.B., L.Y., I.A.G. J.S.C., D.L. and J.M.; Investigation, G.B., L.Y., D.L. and J.M.; Data analysis and interpretation, G.B., L.Y., T.I.R., D.L., J.S.C. and J.M.; Writing—Original Draft, G.B. and J.M.; Writing—Review & Editing, G.B., L.Y., T.I.R., J.S.C., D.L. and J.M.; Funding Acquisition J.S.C., D.L. and J.M.; Supervision, J.S.C. and J.M.

## Additional information

**Competing interests:** The authors declare no competing interests.

