## [Peer Review File · Nature Communications]

Reviewers' comments:

Reviewer #1 (Remarks to the Author):

Despite many advances in the methods for analyzing protein ubiquitylation, mapping of ubiquitin ligase substrates remains a challenging task. In this manuscript Bakos et al. present a new method for identification of RING-type ubiquitin E3 ligase substrates. In this method, the in vitro generated biotinylated E2-ubiquitin thioester is incubated with protein extracts in which endogenous E1/E2 enzymes are inactivated, and in the presence or absence of E3 ligase of interest. Biotinylated proteins from different extracts are then purified using Streptavidin and analyzed by quantitative mass spectrometry.

This is a neat approach for mapping ubiquitin ligase-substrate relations that has an advantage that it is relatively simple and can be easily applied to different cells, tissues and organisms. As the authors demonstrate it could be also used to analyze substrates of SUMO E3 ligases.

The main disadvantage of this method is the fact that E2s used for ubiquitylation of proteins in extracts are charged with biotinylated ubiquitin in vitro and therefore removed out of its cellular context. This raises the question whether the substrate specificity of the studied E2-E3 pair is retained due to a loss of compartmentalization and regulations through protein-protein interactions and PTMs (this might be very different for different E2s and E3 ligases)?

Another issue is the lack of knowledge about the E2-E3 pairs that function together in protein ubiquitylation in a specific cellular condition, which raises the question which E2s should be used when studying a particular E3 ligase. The relevant E2s are known for only a small number of E3 ligases including the APC/C, whose substrates are analyzed in this manuscript. An alternative would be the usage of a more "promiscuous" E2s such as Ube2Ds that might lead to unspecific ubiquitylation of proteins, which are not physiological substrates of a studied E3 ligase. The authors should test this by extending their analysis to additional ubiquitin ligases.

For this manuscript to be suitable for publication in Nature Communications, the authors would need to show that their approach is applicable to other ubiquitin ligases by extending their analysis from APC/C to additional E2-ubiquitin ligase pairs in human cells. This would make the study more comprehensive and the results would serve as a useful resource for the research community.

Alternatively, the authors could perform a biochemical/functional analysis of the selected novel substrate of APC/C in human cells.

- In Figure 2a the authors claim that the method is E2-specific by showing that UBE2R1 charged with biotinylated ubiquitin cannot ubiquitylate the UBE2C/APC substrate. To claim this, the authors would first need to show that the charged UBE2R1 is functional by showing that it can ubiquitylate its substrates.
- The comparison shown in Venn diagram is too complicated and not informative. This should be done differently or removed to supplement.
- In general the validation of substrates is done again in vitro using the same approach but western blotting instead of MS. The validation of the selected substrates should be done by in vivo ubiquitylation assays e.g. comparing of putative substrate ubiquitylation in mitotic HeLa cells in wild type and APC/C, UBE2C and UBE2S knockdown conditions (either from cells expressing tagged Ubiquitin or using tagged substrate proteins).

Reviewer #2 (Remarks to the Author):

Signal transduction by ubiquitin involves the regulation of thousands of substrates. The ubiquitination machinery is highly complex, containing over 600 different E3 ligases and tens of E2. Delineating the substrates regulated by specific E2-E3 pairs is a daunting task and relevant methodology is still poorly developed. To address this, the authors develop an E2-thioester-driven identification method (E2-dID). They use biotinylated ubiquitin uploaded on a specific E2 enzyme as a thioester conjugate and use wild-type extracts and extracts depleted for a specific E3 ligase,

the APC/C to identify relevant substrates. Substrates conjugated to biotinylated ubiquitin are subsequently purified under denaturing conditions and identified by mass spectrometry.

Key points

1. The authors address a key question in ubiquitin signal transduction and the developed methodology could become widely used in the field. The main drawback of their E2~dID is the use of extracts to carry out ubiquitination assays. All proteins are now freely moving in the extract, in contrast to their normal restricted subcellular localization. Therefore, ubiquitination enzymes can now modify proteins that normally would never be within reach in cells. This could lead to false positive identifications. The authors correctly mention in their discussion the lack of contribution of spatial regulation in their experimental set up, but don't address this. Thorough validation of the new methodology using an orthogonal approach at the endogenous level is thus key to investigate the reliability of their methodology. This could best be done by employing the available antibody directed against diglycines attached to lysines, representing the tryptic remnant of ubiquitin as published by the groups of Steve Gygi, Steve Elledge and Wade Harper (Emanuele et al. 2011 Cell and Kim et al. 2011 Mol Cell). In combination with (inducible) knockdown or knockout of the E2 and or E3 under investigation, a quantitative mass spectrometry approach should confirm the reliability of their new methodology. This is key to build trust that E2~dID yields truly reliable substrates and avoids in vitro artefacts. The authors systematically use the term 'in extracto' to avoid the term 'in vitro', but their experiments are clearly carried out in test tubes and therefore qualify as 'in vitro'.

2. The main advantage of the paper is the development of new technology. However, after obtaining a list of new APC/C substrates, the biological aspect of the study remains undeveloped. This is typically the shortcoming of technology driven projects that end with a mass spectrometry list. Identification of new links between the APC/C and the CCR4-NOT deadenylase complex and nonsense-mediated mRNA decay provide excellent opportunities for exciting new biology, but this remains completely untouched unfortunately. The experience of the corresponding author obtained in the lab of Jonathan Pines should enable him to pursue the novel biological aspects of APC/C signalling in a much more substantial manner.

3. The APC/C employs two E2 enzymes, UBE2C and UBE2S. UBE2C is the E2 involved in the first ubiquitination events, whereas UBE2S carries out chain extension. In the current project, only UBE2C is loaded with biotinylated ubiquitin. Employing both E2 enzymes loaded with biotinylated ubiquitin could boost the identification of new APC/C substrates significantly.

Other points

4. Anti APC4 antibody is used to immunodeplete the APC/C. Whereas immunodepletion of APC4 is convincing, it is unclear whether the many other subunits of the APC/C are co-depleted. Thorough immunoblotting for all subunits should be carried out to address this.

5. The authors use a UBE2C-K119R mutant to limit auto-modification. I don't have a problem with that, but I don't accept labelling this mutant simply as UBE2C in further experiments. The mutant should consequently be labelled as UBE2C-K119R.

6. A ~25-fold excess of recombinant UBE2C loaded with biotinylated ubiquitin is used, but these data are not shown. It would be important to show the relative excess by immunoblotting.

7. Labelling a two-fold enrichment as 'stringent thresholds' is overselling.

8. 'but may also provide new means to target ubiquitin and UBL systems in therapy' at the end of the discussion section needs to be removed since it remains completely unclear and undiscussed

how this could be achieved.

9. Adapting the technique to SMT3 is interesting, but the number of identified targets is fairly limited. I would expect SMT3 to have substantially more substrates driven by SIZ1/2. As a quality control, it is common to verify whether the identified targets are predominantly nuclear in agreement with the subcellular localization of SMT3. This needs to be added to ensure that the obtained data are not merely in vitro artefacts.

10. GlyGly sites for ubiquitin are included as modifications in the mass spec search, yet in the main text, it is unclear how many ubiquitin sites are found by mass spec.

11. GlyGly sites for SMT3 are included as modifications in the mass spec search. This is rather surprising since trypsin digestion leaves a considerably longer SMT3 remnant attached to its target. Furthermore, in the main text it is unclear how many SMT3 sites are identified by mass spec.

Reviewer #3 (Remarks to the Author):

The present report describes E2~dID: a method that uses E2-bioUB conjugates, produced in vitro with a multi-step protocol, to help identify potential substrates of an E3 ligase associated with that particular E2. As a test case, the E2-E3 pair UBE2C-APC/C was examined using this method on HeLa cell extracts, with successful identification of known and new substrates for this well-studied cell cycle regulator. To demonstrate that the method was applicable to other ubiquitin-like proteins in other models, known and new substrates of yeast SUMO ligases Siz1/Siz2 were identified. Validation and further characterization of selected substrates is also provided.

Data provided is high quality and reporting of methods is very thorough, especially with details of mass spectrometry. The method is a novel variation on use of biotinylated Ub or Ubls to fish for substrates, to ask more specific questions regarding particular E2/E3 pairs. New substrates are presented, which extend the reach somewhat beyond a "methods" report; the test cases are well-studied E2/E3 combinations and therefore new substrates suggests the method is more sensitive or accesses new proteins due to nature of the method. E2~dID has general interest for labs working in Ub/Ub-likes, especially those with expertise in protein biochemistry, but may be more suited for a biochemical journal. Test cases provided are well-studied E2s and E3s, so versatility for less-studied E2/E3 combinations is questionable. Both test cases involve cell lysates (HeLa or yeast) which may lead to cleaner results and more soluble proteins. One wonders how E2-dID would work with tissue/organ lysates or samples of limited quantity.

General comments:

Authors claim that the method is versatile, which is partially true since it can be applied to most likely all Ub/Ubls and E2s. However, several points bring this versatility into question:

The method depends on having a battery of recombinant and/or in vitro expressed proteins (some of which are available commercially). Although very well described, it will likely be limited to labs with strength in protein biochemistry. Use of dye-labelling/detection of substrates during setup/validation also requires fluo-detection for gels/blots.

E2-dID method is contrasted as being easier than time-consuming in vivo biotinylation methods. Even if they address different questions, they seem both challenging to set up, depending on the expertise of a particular lab.

The ability to link a particular E2 to a particular E3 depends on previously published data on this specificity, which is available in the case of APC/C. As mentioned, E2s can be sometimes used by various E3s, so the availability of specific (and abundant) antibodies for purification/depletion analyses (done with ANAPC4 or APC4 antibodies in this case) is necessary to confirm which E3 is actually responsible. One wonders how easy it would be to apply this approach to a less-studied

ubiquitin E3 ligase that lacks background literature, a well-defined matching E2, and good IP-compatible antibodies.

The method is carried out in cell extracts. The authors mention one drawback is that spatial regulation is lost, but no mention is made of loss of potential substrates due to poor solubility in the buffer conditions used.

The method uses recombinant Ubs and E2s. The authors realize that their test E2 (UBE2C) undergoes auto-ubiquitination, and therefore switch to a K119R mutant for the rest of experiments. Yeast Ubc9 is also reported to undergo SUMOylation (PMID 23644018), but it appears this was not seen or checked by authors. Therefore, commercial E2s may or may not work, and one may require mutant forms to get the method to work. Also, post-translational modifications of Ub/Ubls and E2s will not be added to recombinant proteins, and may/may not occur in extracts with conditions used. Examples include Ub phosphorylation (PMID 26839319), SUMO and UBC9 acetylation (PMID 22578841, 23395904). Although it's likely to influence results in a minor way, it should be acknowledged.

A minor drawback is that HECT and RBR E3 ligases are excluded from analysis by E2-dID, due to need for chemical inhibition via iodoacetamide. A workaround for this is suggested in Discussion, but requires more biochemical steps and production of the E3 ligases themselves (often large and difficult to express).

Authors should address the pervasive use of iodoacetamide (rather than chloroacetamide) throughout the work, in light of a previous publication suggesting that it can produce lysine adducts that mimic the ubiquitin signature diglycine tag (PMID 18511913). While the focus of the paper is not to identify modification sites per se, mention should be made to alert potential users that might want to use E2-dID for this purpose.

It is hard to judge the benchmarking analysis to compare their results with previous results, since mass spectrometry instrumentation is advancing at a rapid pace, and this study uses a very sensitive, latest generation setup. This will certainly influence the number/quality of IDs, so side-by-side comparisons with older data is challenging.

Methods:

Use of ZM447439 (Aurora kinase inhibitor) is specified in the preparation of HeLa K extracts, whereas reversine was used for CDC20-enriched extracts, and seeming neither was used for FZR1-enriched extracts.

Please describe need for these inhibitors, why different inhibitors were used for HeLa K vs. HeLa K/CDC20-enriched experiments, their omission from the FZR1 experiments, and potential influence on results.

"Nitrogen cavitation" as a method for preparing cell lysates is less common and should be referenced. Also, equipment used should be stated.

Mass spectrometry methods and analysis is very well-described. Statistics are correct and threshold cutoffs for positive candidates are reasonable and explained.

Note: Reference to UFM1 samples should be removed from methods, since no UFM1-related samples are reported in this paper.

....the above GlyGly (GG) replaced by ValGly(VG) for UFM1 (Ubiquitin-fold modifier 1) samples. Peptides were validated....

Figures:

Fig2C: Arrowhead is placed in lane 4, exactly where a band should be absent... although it's clear that no band is behind, it's a strange placement.

Fig 5B: Lack of or reduction of SUMOylated bands in Siz1/Siz2 mutants is not entirely convincing when considering the quantity of unmodified. Quantitation should be shown for these bands, although burnout of lane 1 anti-HA will make it difficult.

Supplements:

The supplementary tables have strange names (e.g. 157188_0_data_set_2780846_p3h50t.xlsx). Perhaps this name is assigned by online submission system. They should be renamed with labels from text (ie Supp Table 1), appearing in file title and/or within the table to facilitate identification for reviewers/readers. Detailed description of plasmids/oligos/antibodies is provided.

Point-by-point response to the reviewers

We would like to thank the reviewers for their fair and very useful comments, which we believe have considerably improved our manuscript. We hope that you now agree that our manuscript is suitable for being published in Nature Communications. Please find our point-by-point response below.

Reviewer #1 (Remarks to the Author):

1) The main disadvantage of this method is the fact that E2s used for ubiquitylation of proteins in extracts are charged with biotinylated ubiquitin *in vitro* and therefore removed out of its cellular context. This raises the question whether the substrate specificity of the studied E2-E3 pair is retained due to a loss of compartmentalization and regulations through protein-protein interactions and PTMs (this might be very different for different E2s and E3 ligases)?

This is an important point and we have addressed this issue from two different angles. First, we performed E2~dID from mitotic cells, a cell cycle stage where already a comprehensive list of APC/C substrates is known and characterized *in vivo*. This enabled us compiling a reference list of experimentally-confirmed APC/C substrates to assess how many of these substrates E2~dID correctly can predict. We find that more than half of these substrates are ubiquitinated by UBE2C~bioUBB thioesters in an APC/C-dependent manner suggesting that indeed substrate specificity *is* maintained during E2~dID (Supplementary Figure 3b and Supplementary Table 2). We also show for CCNB1 that only the APC/C-compatible E2 enzymes UBE2C and UBE2D1, but not the SCF E2 enzyme UBE2R1 can ubiquitinate CCNB1, always in a strictly APC/C-dependent manner. Hence, at least in these cases E2~dID recapitulates the situation *in vivo* (Figure 2a-c and Supplementary Figure 2h).

Second, in our revisions we have performed quantitative diGly proteomics based on diGly enrichment and TMT labelling from cells, where the APC/C has been inactivated specifically in mitosis by depleting the ANAPC4 subunit with an auxin-dependent nanobody. We then assessed whether or not the ubiquitination state of candidates predicted by E2~dID is sensitive to APC/C interference as a measure of substrate specificity. Indeed, we find that the ubiquitination of substrates identified by E2~dID is strongly decreased by APC/C interference suggesting that they are targets of the APC/C *in vivo* as well (Fig. 3e and Supplementary Table 3). Analyzing the diGly data we noted however, that the diGly data set likely represent a proteomic snapshot recapitulating an early anaphase stage just prior to the ubiquitination and degradation of Aurora A and B kinases (Figure 4b). In contrast, the substrates suggested by E2~dID cover a much wider time window starting from prometaphase substrates (CCNA2, NEK2A), metaphase (CCNB1, PTTG1, GMNN), anaphase (AURKA, AURKB) to G1 phase (TK)(Fig. 3a). This is not necessarily a disadvantage, because thereby a much wider substrate range can be sampled by E2~dID, while still maintaining substrate specificity. (see also response to reviewer 2, point 1)

Validating 6 E2~dID candidates, which so far have not been associated with the APC/C in more detail (Figs. 4 and 5, Supplementary Figures 4 and 5) we provide in our revision evidence that 33 of 57 E2~dID candidates (excluding the 3 identified APC/C subunits) are substrates of the APC/C *in vivo*. For two further substrates (UPF3B and LSM14B), we provide evidence that they can be ubiquitinated *in vitro* by the APC/C in an APC/C-degron (KEN box)-dependent manner (Fig. 4 and Supplementary Figure 5). Thus, with an *in vivo* “hit quote” of at least ~57% E2~dID performs well in identifying *in vivo* APC/C substrates, also in comparison to alternative approaches (Supplementary Figure 3b).

Nevertheless, we highlight at two places in our manuscript that E2~dID cannot recapitulate all aspects of *in vivo* ubiquitination:

Results, p.10, bottom paragraph:

“During E2~dID ubiquitination of proteins occurs in vitro in the context of total extracts, which do not recapitulate all aspects of substrate ubiquitination such as compartmentalization in vivo. Further, during extract preparation not all proteins will be retained in the soluble fraction.”

Discussion, p.18, top paragraph:

“Extracts may not recapitulate all features required for the faithful interaction of an E3 ligase with its substrates, e.g. the contribution of spatial regulation. Nevertheless, key characteristics of the source material such as a particular cell cycle phase, a differentiation stage or tissue-specificity are retained in the extract and will contribute to E3 selectivity and specificity. Indeed, diGly proteomics from mitotic cells with and without APC/C activity suggest that ubiquitination of canonical and E2~dID-predicted APC/C substrates such as SGO2 and DEPDC1 depend on the APC/C in vivo (Figs. 3, 4).”

Finally, as we have primarily focused on the APC/C as a proof of principle, we cannot exclude that for other E2-E3 pairings specificity is not as strict. However, as in most cases substrate specificity strictly depends on the E3 ligase, we predict that as long the targeted E3 can be efficiently inactivated for E2~dID, substrate specificity will be also kept *in vitro*. In agreement, even when the promiscuous E2 enzyme UBE2D1 is used for E2~dID with the APC/C, ubiquitination of CCNB1 still required the presence of the APC/C (Supplementary Figure 2h).

2) Another issue is the lack of knowledge about the E2-E3 pairs that function together in protein ubiquitylation in a specific cellular condition, which raises the question which E2s should be used when studying a particular E3 ligase. The relevant E2s are known for only a small number of E3 ligases including the APC/C, whose substrates are analyzed in this manuscript. An alternative would be the usage of a more “promiscuous” E2s such as Ube2Ds that might lead to unspecific ubiquitylation of proteins, which are not physiological substrates of a studied E3 ligase. The authors should test this by extending their analysis to additional ubiquitin ligases.

The reviewer raises an excellent point here and provides also a solution to the problem. As described above, we show in Supplementary Figure 2h that

ubiquitination of APC/C substrates driven by promiscuous UBE2D1~bioUBB thioesters remains strictly APC/C dependent. Unlike for UBE2C~bioUBB driven ubiquitination, depleting the ANAPC4 did not strongly reduce the total ubiquitination of proteins modified by UBE2D1~bioUBB. This is in agreement with the notion that UBE2D1 can interact with several E3 ligases. Hence, the use of promiscuous E2 enzymes can be very beneficial as long as the targeted E3 ligases can be inactivated.

3) For this manuscript to be suitable for publication in Nature Communications, the authors would need to show that their approach is applicable to other ubiquitin ligases by extending their analysis from APC/C to additional E2-ubiquitin ligase pairs in human cells. This would make the study more comprehensive and the results would serve as a useful resource for the research community. Alternatively, the authors could perform a biochemical/functional analysis of the selected novel substrate of APC/C in human cells.

As mentioned above we now provide evidence that E2~dID works also with UBE2D1 and UBE2R1 thioesters. Further, we now provide additional data on the verification and characterization of novel APC/C substrates predicted by E2~dID, both *in vitro* and *in vivo*. By performing quantitative diGly proteomics from mitotic cells with or without APC/C activity, we confirm that SGO2, ESPL1 and DEPDC1 are substrates that are also targeted by the APC/C *in vivo* (Fig. 3, 4). We show that SGO2 is degraded during mitotic exit, and DEPDC1 has recently been verified as an APC/C^{FZR1} substrate that is degraded during this time window as well (10.1016/j.yexcr.2017.06.005). Further, we show that SGO2 and ESPL1 interact predominately with the mitotic but not the interphase APC/C, in agreement with the time of their ubiquitination (Fig. 4d and Supplementary Figure 4a, b). Finally, we provide additional *in vitro* characterization LSM14B, showing that its ubiquitin chains can be elongated by UBE2S, and that its ubiquitination by the APC/C depends on a KEN motif.

4) In Figure 2a the authors claim that the method is E2-specific by showing that UBE2R1 charged with biotinylated ubiquitin cannot ubiquitylate the UBE2C/APC substrate. To claim this, the authors would first need to show that the charged UBE2R1 is functional by showing that it can ubiquitylate its substrates.

We now provide in Supplementary Figure 2d, e evidence showing that UBE2R1 can be charged *in vitro* with bioUBB, and that UBE2R1~bioUBB thioesters can drive the ubiquitination of proteins in the extract during E2~dID. Together, this demonstrates that the inability of UBE2R1 to ubiquitinate CCNB1 is not due to an inactive E2 enzyme or because UBE2R1~bioUBB cannot drive substrate ubiquitination *per se*.

5) The comparison shown in Venn diagram is too complicated and not informative. This should be done differently or removed to supplement.

We agree that a Venn diagram with 5 different data sets is complicated, but nevertheless we think it provides valuable information showing the overlap of different data sets. Especially, for researches that might want to further investigate the APC/C candidates suggested by E2~dID in the future, assessing whether or not the candidate of choice has been suggested by other approaches will be certainly helpful. Nevertheless, we have moved the diagram to the supplement as suggested by the reviewer (Supplementary Figure 3b).

6) In general, the validation of substrates is done again *in vitro* using the same approach but western blotting instead of MS. The validation of the selected substrates should be done by *in vivo* ubiquitylation assays e.g. comparing of putative substrate ubiquitylation in mitotic HeLa cells in wild type and APC/C, UBE2C and UBE2S knockdown conditions (either from cells expressing tagged Ubiquitin or using tagged substrate proteins).

We addressed this point by performing diGly proteomics from mitotic cells in the presence or absence of APC/C activity (see point 1. above). These experiments establish that the ubiquitination of at least SGO2, ESPL1 and DEPDC1 are dependent on the APC/C *in vivo*. We think the diGly proteomics experiment is preferable to overexpressing tagged substrates and assaying their ubiquitination by Western blot because it reveals the ubiquitination of endogenous substrates. Further, we note that the *in vitro* APC/C assays with UPF3B and LSM14B (Fig. 5 and Supplementary Figure 5) represent an orthogonal approach, because they are based on reconstituted APC/C activity assays with purified components and not on extracts as the E2~dID experiments.

Reviewer 2:

1) The authors address a key question in ubiquitin signal transduction and the developed methodology could become widely used in the field. The main drawback of their E2~dID is the use of extracts to carry out ubiquitination assays. All proteins are now freely moving in the extract, in contrast to their normal restricted subcellular localization. Therefore, ubiquitination enzymes can now modify proteins that normally would never be within reach in cells. This could lead to false positive identifications. The authors correctly mention in their discussion the lack of contribution of spatial regulation in their experimental set up, but don't address this. 1) Thorough validation of the new methodology using an orthogonal approach at the endogenous level is thus key to investigate the reliability of their methodology. This could best be done by employing the available antibody directed against diglycines attached to lysines, representing the tryptic remnant of ubiquitin as published by the groups of Steve Gygi, Steve Elledge and Wade Harper (Emanuele et al. 2011 Cell and Kim et al. 2011 Mol Cell). In combination with (inducible) knockdown or knockout of the E2 and or E3 under investigation, a quantitative mass spectrometry approach should confirm the reliability of their new methodology. This is key to build trust that E2~dID yields truly reliable substrates and avoids *in vitro* artefacts.

As suggested by the reviewer, we have performed quantitative diGly proteomics from mitotic cells in the presence and absence of APC/C activity based on TMT labeling and subsequent diGly enrichment with two different diGly-specific antibodies (see also point 1, reviewer 1). We have identified more than 18.000 peptides with diGly signatures and show that overall the abundance of diGly remnants on peptides of E2~dID candidates is highly sensitive to APC/C interference (Fig. 3b), showing the *in vivo* relevance of E2~dID. We note however, that for experimental reasons we needed to implement another mitotic synchronization protocol and thus the proteomic snapshot provided by diGly proteomics does not completely recapitulate the cell cycle stage assayed by E2~dID. The reason for this that during E2~dID, we have depleted the APC/C from anaphase extracts by immuno-depletion. *In vivo*, obtaining anaphase-like extracts in the absence of APC/C activity is only possible by chemical inactivation of CDK1, which forces mitotic exit in the absence of ubiquitylation and degradation. Analyzing the diGly data, we found that the diGly data set likely represents a proteomic snapshot recapitulating an early anaphase stage just prior to the ubiquitination and degradation of Aurora A and B kinases (Fig. 4b). In contrast, the substrates suggested by E2~dID cover a much wider time window starting from prometaphase substrates (CCNA2, NEK2A), metaphase (CCNB1, PTTG1, GMNN), anaphase (AURKA, AURKB) to G1 phase (TK)(Fig. 3a). This is not necessarily a disadvantage, because thereby a much wider substrate range can be sampled by E2~dID, while still maintaining substrate specificity.

2) The authors systematically use the term 'in extracto' to avoid the term 'in vitro', but their experiments are clearly carried out in test tubes and therefore qualify as 'in vitro'.

We prefer to use the word *in extracto*, which has been coined by the Kirschner laboratory. In the context of our manuscript the wording *in extracto* allows us to clearly distinguish the assays performed in extracts from *in vitro* APC/C assays that are based on purified recombinant proteins. To avoid any confusion and clearly distinguish *in extracto* from *in vivo* we have defined "in extracto" on p.4, first paragraph:

"Hence, E2~dID is based on the in vitro production of E2~thioesters containing bioUBB or biotinylated UBL proteins that are able to drive substrate modification in vitro, in the context of total extracts (in extracto)."

3) The main advantage of the paper is the development of new technology. However, after obtaining a list of new APC/C substrates, the biological aspect of the study remains undeveloped. This is typically the shortcoming of technology driven projects that end with a mass spectrometry list. Identification of new links between the APC/C and the CCR4-NOT deadenylase complex and nonsense-mediated mRNA decay provide excellent opportunities for exciting new biology, but this remains completely untouched unfortunately. The experience of the corresponding author obtained in the lab of Jonathan Pines should enable him to pursue the novel biological aspects of APC/C signalling in a much more substantial manner.

See also response to reviewer 1, point 3. In our revisions, we have focussed on providing more *in vivo* evidence on the ubiquitination of E2~dID candidates including SGO2 and ESPL1, which have not been addressed in the first version of the manuscript (Fig 4). We show that SGO2 ubiquitination and degradation in mitosis depends on the APC/C, and that both ESPL1 and SGO2 predominately interact with the mitotic but not the interphase APC/C. Further, we have extended the biochemical characterization of LSM14B and have identified its destruction motif(s) (Fig. 4 and Supplementary Figure 5). Taken together, our study now does not only provide a new technology and a list of potential new APC/C substrates, but goes a step further and characterizes several substrates *in vitro* and/or *in vivo*.

We have also tried to provide more functional data especially on UPF3B and LSM14B. While the mutational analysis *in vitro* strongly suggests that UPF3B and LSM14 are APC/C substrates, we did not find evidence in our experiments that both substrates are targeted for ubiquitin-mediated proteolysis (Fig. 4a). Also fluorescent fusions of LSM14B tagged at either terminus appeared largely stable in HeLa cells (data not shown). Hence, either only a subpopulation of both proteins is targeted for degradation, which we cannot reveal by antibody detection, or their ubiquitination might have non-proteolytic functions. Hence, clearly more in depth investigations that likely require other experimental models than HeLa cells (LSM14B functions predominantly have been described

in meiosis (10.1262/jrd.2017-018), and also the relative levels of UPF3B and its paralog UPF3A are especially crucial for spermatogenesis (10.1016/j.cell.2016.02.046)) are required that we believe beyond the scope of this manuscript.

3) The APC/C employs two E2 enzymes, UBE2C and UBE2S. UBE2C is the E2 involved in the first ubiquitination events, whereas UBE2S carries out chain extension. In the current project, only UBE2C is loaded with biotinylated ubiquitin. Employing both E2 enzymes loaded with biotinylated ubiquitin could boost the identification of new APC/C substrates significantly.

This is an interesting thought and could be beneficial for the few E3 ligases that act together with initiating and elongating E2's including the APC/C. Nevertheless, already the addition of a single bioUBB molecule to a substrate should in principle be sufficient for efficient purification due to the high affinity of the streptavidin-biotin interaction.

4) Anti APC4 antibody is used to immunodeplete the APC/C. Whereas immunodepletion of APC4 is convincing, it is unclear whether the many other subunits of the APC/C are co-depleted. Thorough immunoblotting for all subunits should be carried out to address this

We have addressed this point in Supplementary Figure 2a. While we did not test all APC/C subunits, we show that the catalytically relevant subunits APC2, APC11 and APC10 are efficiently co-depleted with ANAPC4. Further, in Fig. 4c, we show that even when ANAPC4 is depleted to a lesser degree than in our *in vitro* experiments, APC/C substrates such as CCNB1 are strongly stabilized, in agreement with an essential function of ANAPC4 for the APC/C holoenzyme.

5) The authors use a UBE2C-K119R mutant to limit auto-modification. I don't have a problem with that, but I don't accept labelling this mutant simply as UBE2C in further experiments. The mutant should consequently be labelled as UBE2C-K119R

A fair point, we have indicated the use of E2 mutants in all figures and in the text.

6) A ~25-fold excess of recombinant UBE2C loaded with biotinylated ubiquitin is used, but these data are not shown. It would be important to show the relative excess by immunoblotting.

We have performed a titration experiment with recombinant UBE2C and endogenous UBE2C in the context of extracts (Supplementary Figure 2b, c). According to this, we estimate that 1 μ g HeLa cell extract contains ~0.82 ng UBE2C. Hence, while we have used a 24.4-fold excess of UBE2C in the experiments used for mass spectrometry, up to an excess of ~115-fold has been used in E2~dID experiments showing CCNB1 ubiquitination (e.g. Fig. 2b, c). Importantly in all cases E2~dID remained firmly E3 dependent, indicative of a

robust assay. For the relevant mass spectrometry experiments, we have specifically indicated the precise excess of UBE2C in the methods.

7) Labelling a two-fold enrichment as 'stringent thresholds' is overselling.

We have removed the comment from the revised manuscript

8) 'but may also provide new means to target ubiquitin and UBL systems in therapy' at the end of the discussion section needs to be removed since it remains completely unclear and undiscussed how this could be achieved

We have clarified our intend and amended the statement to:

"Hence, E2~dID does not only bear the potential to provide new insights into fundamental cell biological processes, but by virtue of establishing enzyme-substrate relationships may also provide targets within ubiquitin and UBL systems for therapy."

9) Adapting the technique to SMT3 is interesting, but the number of identified targets is fairly limited. I would expect SMT3 to have substantially more substrates driven by SIZ1/2. As a quality control, it is common to verify whether the identified targets are predominantly nuclear in agreement with the subcellular localization of SMT3. This needs to be added to ensure that the obtained data are not merely in vitro artefacts

We have added the subcellular localization of the identified components in the Supplementary Table 5, and indeed, most of the substrates are nuclear proteins. However, localization of the identified factors in the cytoplasm does not necessarily mean that their identification as SUMOylated substrates is an artefact. For example, using our method we identified septins, the most prominent SUMOylated proteins in yeast cells. Septins are not nuclear proteins, but localize at the bud neck (Johnson and Gupta (2001), Cell 106, 735-744). In addition, SUMO and Siz1 are exported from the nucleus and can also be found at the bud neck in yeast cells (Makhnevych, et al., (2007). J. Cell Biol. 177, 39-49.).

10) GlyGly sites for ubiquitin are included as modifications in the mass spec search, yet in the main text, it is unclear how many ubiquitin sites are found by mass spec

As the remnant GG modification contains a primary amine, a reaction with the TMT reagents will occur and the mass shift will be $114.0430(\text{GG}) + 229.1629(\text{TMT6plex}) = 343.2059(\text{GGTMT6plex})$. We have searched the E2~dID data (APC/C experiment) with this user defined modification and we identified 155 ubiquitinated peptides. A subset of 27 ubiquitinated peptides were matched to the proposed E2~dID candidate substrates. Since no enrichment for ubiquitinated peptides (but for ubiquitinated proteins) was performed in this instance, the coverage is low and provides incomplete information that can not lead to clear conclusions. For this reason, we do not include this information in the main text. Instead we have included

diGly proteomics as a new validation experiment with deep coverage of ubiquitinated peptides using immunoaffinity enrichment (Fig. 3b-e and Supplementary Table 3).

11) GlyGly sites for SMT3 are included as modifications in the mass spec search. This is rather surprising since trypsin digestion leaves a considerably longer SMT3 remnant attached to its target. Furthermore, in the main text it is unclear how many SMT3 sites are identified by mass spec

In our experiment, we have used a SMT3 mutant where the Ile before the diGly of SMT3 has been exchanged to Lys according to (doi: 10.1038/nprot.2015.095). This would in principle enable diGly enrichment as for ubiquitin and thus after digestion a diGly and not the longer SMT3 remnant is expected. However, since we did not aim to primarily identify SUMOylated peptides in this experiment, we performed trypsinization and not Lys-C digestion. Thus, we cannot distinguish between diGly remnants from ubiquitin and SUMO and therefore no info on SMT3 sites is given in the main text. The information on the mutant has now been added to the methods and Supplementary Table 6.

Reviewer 3:

New substrates are presented, which extend the reach somewhat beyond a “methods” report; the test cases are well-studied E2/E3 combinations and therefore new substrates suggests the method is more sensitive or accesses new proteins due to nature of the method. E2~dID has general interest for labs working in Ub/Ub-likes, especially those with expertise in protein biochemistry, but may be more suited for a biochemical journal.

1) Test cases provided are well-studied E2s and E3s, so versatility for less-studied E2/E3 combinations is questionable.

Due to the high degree of conservation of the enzymology within the ubiquitin system, we expect E2~dID to work in principle with most, if not all RING-type E3 ligases. In our study, we provide evidence of successful E2~dID with two different E2 enzymes and the APC/C, the probably most complex E3 known thus far. Also, E2~dID with UBE2R1 resulted in efficient attachment of bioUBB to proteins, suggesting that our method should also work with SCF-like E3s. Finally, in context of a parallel study, we have successfully performed E2~dID with the ubiquitin-like modifier UFM1 (Hence, the unnecessary reference to UFM1 in the method section this reviewer correctly pointed out).

2) Both test cases involve cell lysates (HeLa or yeast) which may lead to cleaner results and more soluble proteins. One wonders how E2-dID would work with tissue/organ lysates or samples of limited quantity.

A fair point. Clearly the quality of the extract will contribute to the performance of E2~dID. We have not yet tested tissue samples, however the E2~dID experiment for UFM1 mentioned above was performed in total crude extracts, which not have been cleared as UFL1, the only known UFM1 ligating E3, is potentially membrane-bound. In our validation experiments shown in Fig. 2, E2~dID was performed with 1 mg of extract and yielded a good signal in WB analysis. Hence, with a sensitive mass spectrometry setup, we expect E2~dID also to work with samples of more limited quantity as those we have used.

3) Authors claim that the method is versatile, which is partially true since it can be applied to most likely all Ub/Ubls and E2s. However, several points bring this versatility into question:

The method depends on having a battery of recombinant and/or in vitro expressed proteins (some of which are available commercially). Although very well described, it will likely be limited to labs with strength in protein biochemistry. Use of dye-labelling/detection of substrates during setup/validation also requires fluo-detection for gels/blots. E2-dID method is contrasted as being easier than time-consuming in vivo biotinylation methods. Even if they address different questions, they seem both challenging to set up, depending on the expertise of a particular lab.

Currently, more than 80% (35) of the known E2 enzymes in human cells and all ubiquitin-like modifiers are commercially available (e.g. Boston Biochem). The same holds true for BIR A. Hence, except for a few E2's all recombinant components needed for E2~dID are commercially available and should enable also laboratories with limited biochemical experience to successfully perform E2~dID.

The subsequent analysis of E2~dID hits most likely will strongly be guided by the experience of the E2~dID-performing laboratory. However, experiments to setup E2~dID only require SDS-PAGE analysis to assess E2~bioUBB thioester formation and detection of a known model substrate of the targeted E3. While this can be done by dye-labelling (fast, no Western blot required), we also used two alternative methods to validate ubiquitination of substrates in our study. First, simple Western blot detection or S^{35} -labelled *in vitro* translation using a commercially available kit (e.g. Fig. 5). Finally, compared to the average protein, expressing ubiquitin and E2 enzymes is relatively easy (good folding, high yield) and thus should also enable laboratories with less financial support to perform E2~dID.

4) The ability to link a particular E2 to a particular E3 depends on previously published data on this specificity, which is available in the case of APC/C. As mentioned, E2s can be sometimes used by various E3s, so the availability of specific (and abundant) antibodies for purification/depletion analyses (done with ANAPC4 or APC4 antibodies in this case) is necessary to confirm which E3 is actually responsible. One wonders how easy it would be to apply this approach to a less-studied ubiquitin E3 ligase that lacks background literature, a well-defined matching E2, and good IP-compatible antibodies.

The interference with the E3 of choice can be done in different ways. On our revised study we present three different possibilities: 1) Antibody depletion (ANAPC4), 2) genetic inactivation (Siz1/Siz2) and 3) auxin-mediated protein degradation (ANAPC4) in living cells. Another very accessible approach are the increasing numbers of inhibitors. In the light that more and more good antibodies are commercially available and that custom antibody generation is still a key step in characterizing especially less-studied E3's, we envision that antibody-based E3 depletion will be best suited to most users.

Using the promiscuous E2 enzyme UBE2D1 with APC/C, we show that not necessarily a well-defined and specific E2-E3 pairing as UBE2C-APC/C is required for successful E2~dID (see also comment to reviewer 1, point 2).

5) The method is carried out in cell extracts. The authors mention one drawback is that spatial regulation is lost, but no mention is made of loss of potential substrates due to poor solubility in the buffer conditions used.

We have added this information to the main text, p.10 bottom paragraph:

“During E2~dID ubiquitination of proteins occurs in vitro in the context of total extracts, which do not recapitulate all aspects of substrate ubiquitination such as compartmentalization in vivo. Further, during extract preparation not all proteins will be retained in the soluble fraction.”

Of note, as indicated above (point 2), we have evidence that E2~dID can also be carried out in crude extracts.

6) The method uses recombinant Ubs and E2s. The authors realize that their test E2 (UBE2C) undergoes auto-ubiquitination, and therefore switch to a K119R mutant for the rest of experiments. Yeast Ubc9 is also reported to undergo SUMOylation (PMID 23644018), but it appears this was not seen or checked by authors.

In fact, we generated the UBC9^{K153R/K157R} mutant described in PMID 23644018 before we attempted performing E2~dID with SUMO. However, in our hands we did not see a significant decrease in autoSUMOylation of UBC9^{K153R/K157R}, and therefore we chose to use WT UBC9.

7) Therefore, commercial E2s may or may not work, and one may require mutant forms to get the method to work.

Auto-ubiquitination/modification strongly differs from E2 to E2. For example, while UBE2C and UBC9 automodify themselves, neither UBE2D1 nor UBE2R1 show significant auto-ubiquitination (Supplementary Figure 2). Thus, it is difficult to predict how much of a problem auto-ubiquitination will be for all E2 enzymes. Nevertheless, our experiments with UBC9 indicate that E2~dID can be performed even with an E2 that has a tendency to autoubiquitinate itself.

8) Also, post-translational modifications of Ub/UBLs and E2s will not be added to recombinant proteins, and may/may not occur in extracts with conditions used. Examples include Ub phosphorylation (PMID 26839319), SUMO and UBC9 acetylation (PMID 22578841, 23395904). Although it's likely to influence results in a minor way, it should be acknowledged.

In the revised manuscript we now discuss this potential issue on p.18, bottom paragraph:

“A potential caveat of using recombinant E2's, ubiquitin and UBLs is their lack of posttranslational modifications that might contribute to function. For example, Ser65-phosphorylation of ubiquitin activates the PARKIN E3 ubiquitin ligase (10.1038/nature14879) and acetylation of SUMO and UBC9 regulate SUMO-mediated interactions (10.1016/j.molcel.2012.04.006) and substrate binding (10.1038/emboj.2013.5), respectively. Nevertheless, it is possible that extracts are capable

of correctly modifying the supplied recombinant protein as they are also able to drive ubiquitination using the in the extract present E3 enzymes.”

9) A minor drawback is that HECT and RBR E3 ligases are excluded from analysis by E2-dID, due to need for chemical inhibition via iodoacetamide. A workaround for this is suggested in Discussion, but requires more biochemical steps and production of the E3 ligases themselves (often large and difficult to express)

This is true and as mentioned by the reviewer is already highlighted in the discussion. We note however, that from all known E3 ligases HECT and RBR E3 ligases represent a relatively small group and thus E2~dID can be in principle employed with the majority of all E3 ligases.

10) The Authors should address the pervasive use of iodoacetamide (rather than chloroacetamide) throughout the work, in light of a previous publication suggesting that it can produce lysine adducts that mimic the ubiquitin signature diglycine tag (PMID 18511913). While the focus of the paper is not to identify modification sites per se, mention should be made to alert potential users that might want to use E2-dID for this purpose.

This is not a problem in TMT experiments of ubiquitinated peptides. As the remnant GG modification contains a primary amine, a reaction with the TMT reagents will occur and the mass shift will be $114.0430(\text{GG}) + 229.1629(\text{TMT6plex}) = 343.2059(\text{GGTMT6plex})$; which is very different from the mass of the lysine adducts. The chemical modification of the remnant GG by TMT is also the reason why the KGG antibodies do not work in already TMT labelled peptides (also mentioned here: <https://doi.org/10.1016/j.cels.2016.08.009>)

Nevertheless, to make sure that in alternative mass spectrometry strategies the use of iodoacetamide does not lead to misidentification of ubiquitin sites we have included this statement into the discussion, p19, top paragraph:

“In our experiments, we have used IAA because in vitro the APC/C appears to be sensitive to NEM-treatment (Supplementary Figure 2). IAA can produce lysine adducts that mimic the ubiquitin signature diglycine tag{Nielsen:2008dm}. This is not an issue in the context of the TMT labelling experiments we employed for E2~dID and diGly proteomics, because after trypsin digestion the remnant diGly modification contains a primary amine that will react with the TMT label thus leading to a mass shift, which is very different from the mass of the lysine adducts. Nevertheless, in different experimental workflows alternative alkylating reagents such as chloroacetamide or bromoacetamide should be considered to avoid misidentification of ubiquitin sites.”

11) It is hard to judge the benchmarking analysis to compare their results with previous results, since mass spectrometry instrumentation is advancing at a rapid pace, and this study uses a very sensitive, latest generation setup. This will certainly influence the number/quality of IDs, so side-by-side comparisons with older data is challenging.

We added this sentence to acknowledge this point (p21, top paragraph):

“Based on the number of identified reference substrates E2~dID performs also well in predicting bonafide APC/C substrates compared to recent indirect approaches, such as the mitotic exit ubiquitome {Min:2014da} or co-regulation proteomics {Singh:2014ih}. While our study certainly benefited from the ever-increasing sensitivity in mass spectrometry approaches, this is not surprising since both approaches solely depend on the activity profile of the APC/C during mitosis to enrich its substrates.”

We note that main difference of E2~dID compared to the two other approaches that employed mass spectrometry (co-regulation proteomics and the mitotic exit ubiquitinome) is that:

- 1) E2~dID directly identifies the modified substrate in a E3 ligase-dependent manner.
- 2) For E2~dID the fate of the ubiquitinated substrate (e.g. degradation) does not matter (unlike for co-regulation proteomics).

12) Methods:

Use of ZM447439 (Aurora kinase inhibitor) is specified in the preparation of HeLa K extracts, whereas reversine was used for CDC20-enriched extracts, and seeming neither was used for FZR1-enriched extracts.

Please describe need for these inhibitors, why different inhibitors were used for HeLa K vs. HeLa K/CDC20-enriched experiments, their omission from the FZR1 experiments, and potential influence on results.

We use the Aurora inhibitor ZM447439 to achieve a more synchronous release from a prometaphase arrest that is dependent on the mitotic checkpoint. We have included this reasoning into the main text (p12, bottom paragraph):

“Thus, we first monitored the stability of the selected candidates during mitotic exit by releasing taxol-arrested cells from prometaphase in the presence of ZM447439 to increase the synchronicity of the cell population”.

With additional synchronizations protocols required for the revision, we have rewritten and streamlined this section in the methods. The new section now includes more information on the rationale of synchronization and inhibitor use to facilitate application in non-cell cycle laboratories.

13) “Nitrogen cavitation” as a method for preparing cell lysates is less common and should be referenced. Also, equipment used should be stated.

We have added the reference and equipment used

14) Mass spectrometry methods and analysis is very well-described. Statistics are correct and threshold cutoffs for positive candidates are reasonable and explained.

Thank you.

15) Note: Reference to UFM1 samples should be removed from methods, since no UFM1-related samples are reported in this paper.
...the above GlyGly (GG) replaced by ValGly(VG) for UFM1 (Ubiquitin-fold modifier 1) samples. Peptides were validated....

We have indeed performed E2~dID with UFM1 for another study that is not part of this manuscript and overlooked this reference to UFM1. It has been now removed.

Figures:

13) Fig2C: Arrowhead is placed in lane 4, exactly where a band should be absent... although it's clear that no band is behind, it's a strange placement.

We moved the arrowhead accordingly

14) Fig 5B: Lack of or reduction of SUMOylated bands in Siz1/Siz2 mutants is not entirely convincing when considering the quantity of unmodified. Quantitation should be shown for these bands, although burnout of lane 1 anti-HA will make it difficult.

We have repeated the experiments and show now the quantification of the results: for DEF1, the intensity of SUMOylation that is still detectable in the siz1/2 mutant varies between 39 and 6% of the wild-type SUMOylation (variability depending largely on the western background). We are thus confident that deletion of the two SUMO ligases leads to significant decrease in Def1 SUMOylation.

In contrast, we were not able to reproduce the reduction of SUMOylation of Ede1, which surprisingly seems to increase upon deletion of Siz1/2. We suspect that SUMO forms chains on Ede1 that are barely detectable due to the large size of the Ede1 protein (250 kDa); upon deletion of Siz1/2, these chains are replaced by the monoSUMOylated form of the protein that is more readily detectable, giving thus the impression of an increase in SUMOylation. We do not however have any proof of this hypothesis and we thus prefer to retract the data regarding Ede1 from the manuscript.

Supplements:

The supplementary tables have strange names (e.g. 157188_0_data_set_2780846_p3h50t.xlsx). Perhaps this name is assigned by online submission system. They should be renamed with labels from text (ie Supp Table 1), appearing in file title and/or within the table to facilitate identification for reviewers/readers. Detailed description of plasmids/oligos/antibodies is provided.

We have provided Excel Tables named Supplementary Table 1, etc in our previous submission and suspect the changed naming is due to the online submission system.

REVIEWERS' COMMENTS:

Reviewer #1 (Remarks to the Author):

The authors have done a very good job in improving the manuscript. I recommend the publication in Nature Communications after the following minor points are addressed:

There are misspellings/missing words on several places in the text. The authors should carefully read the manuscript and correct this.

I could not access the data in the PRIDE database after logging in with provided username/password: If already not added, the authors should upload the raw MS data from di-glycine lysine IPs. Also, the authors should provide a supplementary table with the results from the same analysis (e.g all quantified di-glycine modified peptides with ratios in ANAPC -/+ extracts).

The authors point out that promiscuous E2s can be used when no information about E2-E3 pairs is available. This is partially true based on their new analysis. However, the authors need to state in the main text that to be able to do this a quantitative approach such as SILAC or TMT needs to be used to distinguish E3-specific substrates from substrates that are modified using the E3 that are present in the extracts and that can pair with the used E2. As shown in Suppl Fig 2g, usage of UBE2Ds will lead to a lot of ubiquitylation even without the E3 ligase, but just using ligases present in the extracts.

In methods Ubiquitinome enrichment is wrong (maybe Ubiquitinome analysis is possible): Di-glycine lysine-modified peptide enrichment or ubiquitin remnant peptide enrichment is correct.

Fig 4a: Should be more clearly labeled what different blots of ESPL1 correspond to.

Fig 4b: Y axis is mislabeled, also what the position of different Ks referees to is unclear

Reviewer #2 (Remarks to the Author):

Most of my comments from the previous round have been addressed by the authors except for my second key point: lack of biological follow-up for new APC/C targets. This is typically the shortcoming of mass spectrometry driven projects that result in a list of proteins. Whereas the authors confirm some new APC/C targets by immunoblotting, the biological effects of ubiquitination of these targets remain unclear. Identification of new APC/C targets provides excellent opportunities for new biology, but this remains largely untouched unfortunately. The authors rule out that ubiquitination affects the protein stability of the total population of some targets, but don't reveal how ubiquitination affects any new target. What are the biological consequences of ubiquitination for these targets? How does it fit in the context of metaphase to anaphase progression? The experience of the corresponding author obtained in the lab of Jonathan Pines should enable him to pursue the novel biological aspects of APC/C signalling in a much more substantial manner.

Point-by-point response to the reviewers

We would like to thank the reviewers for their comments, which are addressed in the point-by-point response below.

Reviewer #1 (Remarks to the Author):

The authors have done a very good job in improving the manuscript. I recommend the publication in Nature Communications after the following minor points are addressed:

There are misspellings/missing words on several places in the text. The authors should carefully read the manuscript and correct this.

We thank the reviewer for the positive evaluation of our manuscript and have corrected misspellings/missing words on several places in the text.

I could not access the data in the PRIDE database after logging in with provided username/password: If already not added, the authors should upload the raw MS data from di-glycine lysine IPs.

All the raw data including the raw MS data from di-glycine lysine IPs have been deposited in the PRIDE database and will be freely available upon publication of this manuscript.

Also, the authors should provide a supplementary table with the results from the same analysis (e.g all quantified di-glycine modified peptides with ratios in ANAPC +/- extracts).

The TMT intensities and ratio calculations of the diGly proteomic experiment with + APC/C and – APC/C extracts are presented as Supplementary Data 3 in the revised manuscript.

The authors point out that promiscuous E2s can be used when no information about E2-E3 pairs is available. This is partially true based on their new analysis. However, the authors need to state in the main text that to be able to do this a quantitative approach such as SILAC or TMT needs to be used to distinguish E3-specific substrates from substrates that are modified using the E3 that are present in the extracts and that can pair with the used E2. As shown in Suppl Fig 2g, usage of UBE2Ds will lead to a lot of ubiquitylation even without the E3 ligase, but just using ligases present in the extracts.

We have included this notion in the discussion: “As substrate specificity depends on E3 ligases, also E2 enzymes that interact with several E3’s can be employed for E2~dID when a quantitative mass spectrometry approach is employed to reveal only substrates of the E3 of interest.”

In methods Ubiquitinome enrichment is wrong (maybe Ubiquitinome analysis is possible): Di-glycine lysine-modified peptide enrichment or ubiquitin remnant peptide enrichment is correct.

We thank the reviewer for this correction and now use the wording “ubiquitin remnant peptide enrichment” as suggested

Fig 4a: Should be more clearly labeled what different blots of ESPL1 correspond to.

We have now clearly labelled, which bands correspond to ESPL1, ESPL1^{N-term} or ESPL1^{C-term}.

Fig 4b: Y axis is mislabeled, also what the position of different Ks referees to is unclear

We have corrected a spelling mistake in the axis labelling. The graph shows the fold-change (log₂) in the abundance of lysine ubiquitination present in single peptides (labelled as K) as indicated on the axis. The K represents a data point (the ratio) of a peptide(s) from each indicated substrate. To avoid misunderstanding we have added additional information in the figure legend:

“Scatter plot showing the fold-change in lysine ubiquitination in response to ANAPC4 depletion for early APC/C substrates and E2~dID candidates selected for further analyses. Each “K” represents the fold-change (log₂) of a ubiquitinated peptide identified from the indicated proteins. The colour gradient from blue (no change) to red (negative change) illustrates the fold-change in the abundance of identified ubiquitinated peptides.

Reviewer #2 (Remarks to the Author):

Most of my comments from the previous round have been addressed by the authors except for my second key point: lack of biological follow-up for new APC/C targets. This is typically the shortcoming of mass spectrometry driven projects that result in a list of proteins. Whereas the authors confirm some new APC/C targets by immunoblotting, the biological effects of ubiquitination of these targets remain unclear. Identification of new APC/C targets provides excellent opportunities for new biology, but this remains largely untouched unfortunately. The authors rule out that ubiquitination affects the protein stability of the total population of some targets, but don't reveal how ubiquitination affects any new target. What are the biological consequences of ubiquitination for these targets? How does it fit in the context of metaphase to anaphase progression? The experience of the corresponding author obtained in the lab of Jonathan Pines should enable him to pursue the novel biological aspects of APC/C signalling in a much more substantial manner.

We thank the reviewer for the overall positive evaluation and politely would like to point out that in our opinion an in depth biological follow-up of new APC/C substrates is beyond the scope of this study. For our method-oriented study, we have rather focussed on providing evidence that several E2~dID candidates spanning the whole candidate ranking are indeed *bonafide* APC/C substrates based on experiments *in vitro* and in living cells. Thereby, we aim to provide trust that E2~dID is also an excellent approach for other scientists with different scientific questions and experimental systems. The reviewer raises several intriguing questions on the biological consequences of the ubiquitination of the APC/C candidates we have identified. Indeed, we aim addressing these questions in a separate study in the future to provide a much more comprehensive and mechanistic analysis than that, what would have been possible in the context of this manuscript.